# Presynaptic NMDA receptors facilitate short-term plasticity and BDNF release at hippocampal mossy fiber synapses

Pablo J Lituma[1], Hyung-Bae Kwon[1†], Karina Alviña[1‡], Rafael Luján[2], Pablo E Castillo[1,3]*

[1]Dominick P. Purpura Department of Neuroscience, Albert Einstein College of Medicine, Bronx, United States; [2]Instituto de Investigación en Discapacidades Neurológicas (IDINE), Facultad de Medicina, Universidad Castilla-La Mancha, Albacete, Spain; [3]Department of Psychiatry and Behavioral Sciences, Albert Einstein College of Medicine, Bronx, United States

**Abstract** Neurotransmitter release is a highly controlled process by which synapses can critically regulate information transfer within neural circuits. While presynaptic receptors – typically activated by neurotransmitters and modulated by neuromodulators – provide a powerful way of fine-tuning synaptic function, their contribution to activity-dependent changes in transmitter release remains poorly understood. Here, we report that presynaptic NMDA receptors (preNMDARs) at mossy fiber boutons in the rodent hippocampus can be activated by physiologically relevant patterns of activity and selectively enhance short-term synaptic plasticity at mossy fiber inputs onto CA3 pyramidal cells and mossy cells, but not onto inhibitory interneurons. Moreover, preNMDARs facilitate brain-derived neurotrophic factor release and contribute to presynaptic calcium rise. Taken together, our results indicate that by increasing presynaptic calcium, preNMDARs fine-tune mossy fiber neurotransmission and can control information transfer during dentate granule cell burst activity that normally occur in vivo.

*For correspondence:
pablo.castillo@einsteinmed.org

Present address: †The Solomon H. Snyder Department of Neuroscience, John Hopkins University, School of Medicine, Baltimore, United States; ‡Department of Neuroscience, University of Florida, Gainesville, United States

Competing interests: The authors declare that no competing interests exist.

## Introduction

Neurotransmission is a dynamic and highly regulated process. The activation of ionotropic and metabotropic presynaptic autoreceptors provides a powerful way of fine-tuning neurotransmission via the facilitation or inhibition of neurotransmitter release (*Burke and Bender, 2019*; *Engelman and MacDermott, 2004*; *Miller, 1998*; *Pinheiro and Mulle, 2008*; *Schicker et al., 2008*). Due to their unique functional properties, including high calcium-permeability, slow kinetics and well-characterized role as coincidence detectors (*Cull-Candy et al., 2001*; *Lau and Zukin, 2007*; *Paoletti et al., 2013*; *Traynelis et al., 2010*), presynaptic NMDA receptors (preNMDARs) have received particular attention (*Banerjee et al., 2016*; *Bouvier et al., 2015*; *Bouvier et al., 2018*; *Duguid, 2013*; *Duguid and Smart, 2009*; *Wong et al., 2021*). Regulation of neurotransmitter release by NMDA autoreceptors in the brain was suggested three decades ago (*Martin et al., 1991*). Anatomical evidence for preNMDARs arose from an immunoelectron microscopy study revealing NMDARs at the mossy fiber giant bouton of the monkey hippocampus (*Siegel et al., 1994*), followed by functional studies in the entorhinal cortex, indicating that preNMDARs tonically increase spontaneous glutamate release and also facilitate evoked release in a frequency-dependent manner (*Berretta and Jones, 1996*; *Woodhall et al., 2001*). Since these early studies, although evidence for preNMDARs has accumulated throughout the brain (*Banerjee et al., 2016*; *Bouvier et al., 2018*; *Duguid and Smart, 2009*), the presence and functional relevance of preNMDARs at key synapses in the brain have been called into question (*Carter and Jahr, 2016*; *Duguid, 2013*).

Mossy fibers (mf) – the axons of dentate granule cells (GCs) – establish excitatory synapses onto proximal dendrites of CA3 pyramidal neurons, thereby conveying a major excitatory input to the hippocampus proper (*Amaral et al., 2007*; *Henze et al., 2000*). This synapse displays uniquely robust frequency facilitation both in vitro (*Nicoll and Schmitz, 2005*; *Salin et al., 1996*; *Vyleta et al., 2016*) and in vivo (*Hagena and Manahan-Vaughan, 2010*; *Vandael et al., 2020*). The molecular basis of this short-term plasticity is not fully understood but likely relies on diverse presynaptic mechanisms that increase glutamate release (*Jackman and Regehr, 2017*; *Rebola et al., 2017*). Short-term, use-dependent facilitation is believed to play a critical role in information transfer, circuit dynamics, and short-term memory (*Abbott and Regehr, 2004*; *Jackman and Regehr, 2017*; *Klug et al., 2012*). The mf-CA3 synapse can strongly drive the CA3 network during short bursts of presynaptic activity (*Chamberland et al., 2018*; *Henze et al., 2002*; *Vyleta et al., 2016*; *Zucca et al., 2017*), an effect that likely results from two key properties of this synapse, namely, its strong frequency facilitation and proximal dendritic localization. In addition to CA3 pyramidal neurons, mf axons establish synaptic connections with hilar mossy cells (MCs) and inhibitory interneurons (INs) (*Amaral et al., 2007*; *Henze et al., 2000*). These connections also display robust short-term plasticity (*Lysetskiy et al., 2005*; *Toth et al., 2000*), which may contribute significantly to information transfer and dynamic modulation of the dentate gyrus (DG)-CA3 circuit (*Bischofberger et al., 2006*; *Evstratova and Tóth, 2014*; *Lawrence and McBain, 2003*). Despite early evidence for preNMDARs at mf boutons (*Siegel et al., 1994*), whether these receptors modulate neurotransmission at mf synapses is unknown. Intriguingly, mfs contain one of the highest expression levels of brain-derived neurotrophic factor, BDNF (*Conner et al., 1997*). While preNMDARs were implicated in BDNF release at corticostriatal synapses (*Park et al., 2014*), whether putative preNMDARs impact BDNF release at mf synapses remains unexplored.

Here, to examine the potential presence and impact of preNMDARs at mf synapses, we utilized multiple approaches, including immunoelectron microscopy, selective pharmacology for NMDARs, a genetic knockout strategy to remove NMDARs from presynaptic GCs, two-photon imaging of BDNF release, and presynaptic $Ca^{2+}$ signals in acute rodent hippocampal slices. Our findings indicate that preNMDARs contribute to mf short-term plasticity and promote BDNF release likely by increasing presynaptic $Ca^{2+}$. Thus, preNMDARs at mfs may facilitate information transfer and provide an important point of regulation in the DG-CA3 circuit by modulating both glutamate and BDNF release.

## Results

### Electron microscopy reveals presynaptic NMDA receptors at mossy fiber terminals

To determine the potential localization of NMDA receptors at the mf terminals of the rodent hippocampus, we performed electron microscopy and post-embedding immunogold labeling in rats using a validated antibody for the obligatory subunit GluN1 (*Petralia et al., 1994*; *Siegel et al., 1994*; *Takumi et al., 1999*; *Watanabe et al., 1998*). Gold particles were detected in the main body of the postsynaptic density as well as presynaptic mf terminals (*Figure 1A–C*). GluN1 localized in mf boutons in a relatively high proportion to the active zone, as compared to associational–commissural (ac) synapse in the same CA3 pyramidal neuron (*Figure 1D*; mf, ~32% presynaptic particles; ac, <10% presynaptic particles; n = 3 animals). Similar quantification for AMPA receptors did not reveal presynaptic localization of these receptors in either mf or ac synapses (*Figure 1—figure supplement 1*; ~5% presynaptic particles, n = 3 animals). Together, these results provide anatomical evidence for preNMDARs at mf-CA3 synapses.

### Both NMDAR antagonism and genetic deletion from presynaptic GCs reduce mossy fiber low-frequency facilitation

Presynaptic short-term plasticity, in the form of low-frequency (~1 Hz) facilitation (LFF), is uniquely robust at the mf-CA3 synapse (*Nicoll and Schmitz, 2005*; *Salin et al., 1996*). To test a potential involvement of preNMDARs in LFF, we monitored AMPAR-mediated excitatory postsynaptic currents (EPSCs) from CA3 pyramidal neurons in acute rat hippocampal slices. Neurons were held at $V_h = -70$ mV to minimize postsynaptic NMDAR conductance, and mfs were focally stimulated with a bipolar electrode (theta glass pipette) placed in *stratum lucidum* ~100 μm from the recorded cell.

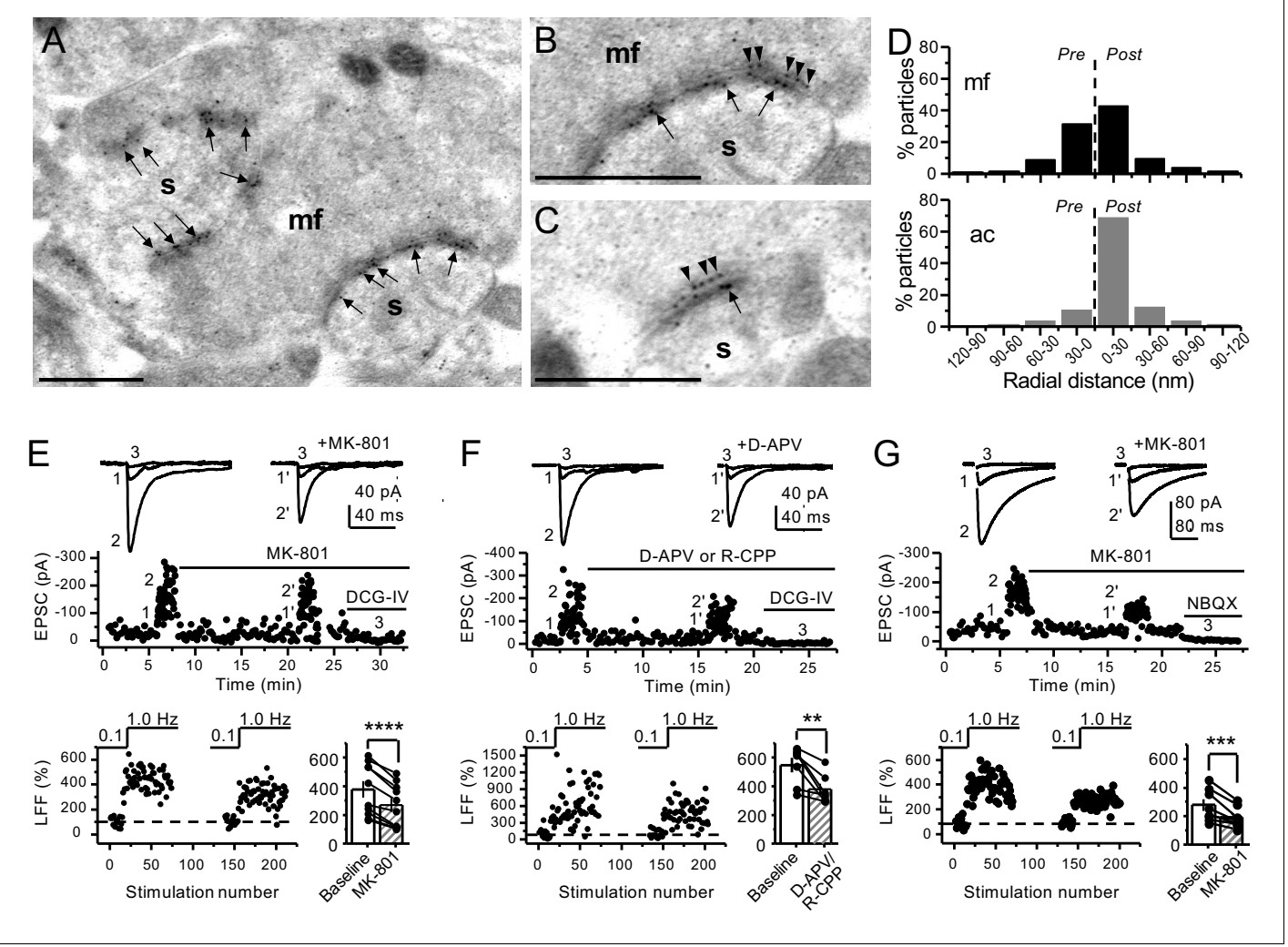

**Figure 1.** Anatomical and functional evidence for preNMDARs at mossy fiber synapses. (A) Image of a mossy fiber (mf) giant bouton and postsynaptic spines (s). (B, C) Higher magnification of mf synapses. Arrows indicate postsynaptic GluN1, whereas arrowheads indicate presynaptic GluN1. Calibration bars: 500 nm. (D) Mossy fiber (mf) and associational–commissural (ac) synaptic GluN1 immuno-particle radial distribution (30 nm bins), mf: 34 synapses, 100 presynaptic particles; ac: 25 synapses, 24 presynaptic particles; three animals. (E) AMPAR-ESPCs were recorded at $V_h$ = −70 mV in the presence of 0.5 μM LY303070 and 100 μM picrotoxin. Low-frequency facilitation (LFF), induced by stepping stimulation frequency from 0.1 to 1 Hz, was assessed before and after bath application of MK-801 (50 μM). MK-801 significantly reduced LFF (baseline 378 ± 57%, MK-801 270 ± 48%, n = 10 cells, nine animals; baseline vs MK-801, p=3.8×10$^{-5}$, paired t-test). In all panels of this figure: representative traces (*top*), representative experiment (*middle*), and normalized LFF and summary plot (*bottom*). DCG-IV (1 μM) was applied at the end of all recordings to confirm mf-CA3 transmission. (F) D-APV (100 μM) or R-CPP (50 μM) application also reduced LFF (baseline 546 ± 50%, D-APV/R-CPP 380 ± 38%, n = 7 cells, five animals; baseline vs D-APV/R-CPP, p=0.00743, paired t-test). (G) KAR-EPSCs were recorded at $V_h$ = −70 mV in the presence of 15 μM LY303070 and 100 μM picrotoxin. In addition, NMDAR-mediated transmission was blocked intracellularly by loading MK-801 (2 mM) in the patch-pipette. Bath application of MK-801 (50 μM) significantly reduced LFF (baseline 278 ± 40%, MK-801 195 ± 26% n = 8 cells, six animals; baseline vs MK-801, p=0.00259, paired t-test). Data are presented as mean ± s.e.m. **p<0.01; ***p<0.005; ****p<0.001.

The online version of this article includes the following figure supplement(s) for figure 1:

**Figure supplement 1.** Immunogold-EM reveals negligible presynaptic AMPAR particle distribution.

**Figure supplement 2.** Stable low-frequency facilitation of mf-CA3 synaptic transmission in naïve slices.

**Figure supplement 3.** Intracellular MK-801 effectively blocked postsynaptic NMDARs.

LFF was induced by stepping the stimulation frequency from 0.1 Hz to 1 Hz for ~2 min in the presence of picrotoxin (100 μM) to block fast inhibitory synaptic transmission, and a low concentration of the AMPAR noncompetitive antagonist LY303070 (0.5 μM) to minimize CA3–CA3 recurrent activity (*Kwon and Castillo, 2008*). Bath application of the NMDAR irreversible open channel blocker MK-

801 (50 µM) significantly reduced LFF (*Figure 1E*). In addition, the competitive NMDAR antagonists D-APV (100 µM) or R-CPP (50 µM) yielded a comparable reduction of facilitation (*Figure 1F*). To confirm that these synaptic responses were mediated by mfs, the mGluR2/3 agonist DCG-IV (1 µM) was applied at the end of all recordings (*Kamiya et al., 1996*). To control for stability, we performed interleaved experiments in the absence of NMDAR antagonists and found that LFF remained unchanged (*Figure 1—figure supplement 2A*). These findings indicate NMDAR antagonism reduces mf-CA3 short-term plasticity (LFF), suggesting that preNMDARs could contribute to this form of presynaptic plasticity.

The reduction in facilitation of AMPAR transmission could be due to dampening of CA3 recurrent activity by NMDAR antagonism (*Henze et al., 2000*; *Kwon and Castillo, 2008*; *Nicoll and Schmitz, 2005*). To discard this possibility, we repeated our experiments in a much less excitable network in which AMPAR-mediated synaptic transmission was selectively blocked by a high concentration of the noncompetitive antagonist LY303070 (15 µM) and monitored the kainate receptor (KAR)-mediated component of mf synaptic transmission (*Castillo et al., 1997*; *Kwon and Castillo, 2008*). In addition, 2 mM MK-801 was included in the intracellular recording solution to block postsynaptic NMDARs (*Corlew et al., 2008*; *Figure 1—figure supplement 3*). To further ensure postsynaptic NMDAR blockade, we voltage-clamped the CA3 pyramidal neuron at −70 mV and waited until NMDAR-mediated transmission was eliminated and only KAR-EPSCs remained. Under these recording conditions, bath application of MK-801 (50 µM) also reduced LFF of KAR-mediated transmission (*Figure 1G*), whereas LFF remained unchanged in interleaved control experiments (*Figure 1—figure supplement 2B*). At the end of these recordings, 10 µM NBQX was applied to confirm KAR transmission (*Figure 1G*, *Figure 1—figure supplement 2B*; *Castillo et al., 1997*; *Kwon and Castillo, 2008*). It is therefore unlikely that the reduction of LFF mediated by NMDAR antagonism could be explained by recurrent network activity, suggesting a direct effect on transmitter release.

To further support a role of preNMDARs in mf LFF, we took a genetic approach by conditionally removing NMDARs from GCs in *Grin1* floxed mice. To this end, an AAV5-CaMKII-Cre-GFP virus was bilaterally injected in the DG to selectively delete *Grin1* , whereas AAV5-CaMKII-eGFP was injected in littermates as a control at postnatal days 16–20 in both groups (*Figure 2A*). Two weeks after surgery, we prepared acute hippocampal slices and examined the efficacy of *Grin1* deletion by analyzing NMDAR-mediated transmission in GFP$^+$ GCs of *Grin1*-cKO and control mice. We confirmed that in contrast to control mice, no NMDAR-EPSCs were elicited by electrically stimulating medial perforant-path inputs in *Grin1*-cKO GCs voltage-clamped at +40 mV in the presence of 100 µM picrotoxin and 10 µM NBQX (*Figure 2B*). As expected, the NMDAR/AMPAR ratio was significantly reduced in *Grin1*-cKO mice compared to control (*Figure 2C*). Only acute slices that exhibited robust GFP fluorescence in the DG were tested for LFF of AMPAR transmission in CA3. We found that LFF was significantly reduced in *Grin1*-cKOs as compared to controls (*Figure 2D*), indicating that genetic removal of NMDARs from GCs recapitulated NMDAR antagonism (*Figure 1E–G*). *Grin1* deletion did not affect basal transmitter release as indicated by a comparable paired-pulse ratio to control (Control: 2.5 ± 0.36, n = 13 cells; *Grin1* cKO: 2.4 ± 0.31, n = 13 cells; U > 0.5, Mann–Whitney test). Collectively, our findings using two distinct approaches strongly suggest that NMDAR activation in GCs increases LFF of mf-CA3 synaptic transmission.

## Reduced facilitation by NMDAR antagonism is independent of the GC somatodendritic compartment

Bath application of MK-801 could have blocked dendritic NMDARs in GCs and potentially affected transmitter release (*Christie and Jahr, 2008*; *Duguid, 2013*). To address this possibility, we repeated our experiments after performing a surgical cut in the granular layer of the DG in order to isolate mf axons from GCs (*Figure 3—figure supplement 1A*). Under these conditions, MK-801 bath application still reduced LFF (*Figure 3A*), and LFF was stable in control, acutely transected axons (*Figure 3B*). In addition, puffing D-APV (2 mM) in *stratum lucidum* near (~200 µm) the recorded neuron also reduced LFF (*Figure 3C*), whereas puffing artificial cerebrospinal fluid (ACSF) had no effect (*Figure 3D*). Lastly, in a set of control experiments, we confirmed that D-APV puffs were sufficient to transiently block NMDAR-mediated transmission in CA3, but not in DG (*Figure 3—figure supplement 1B,C*). Together, these results support the notion that LFF reduction was due to the blockade of preNMDARs but not somatodendritic NMDARs on GCs.

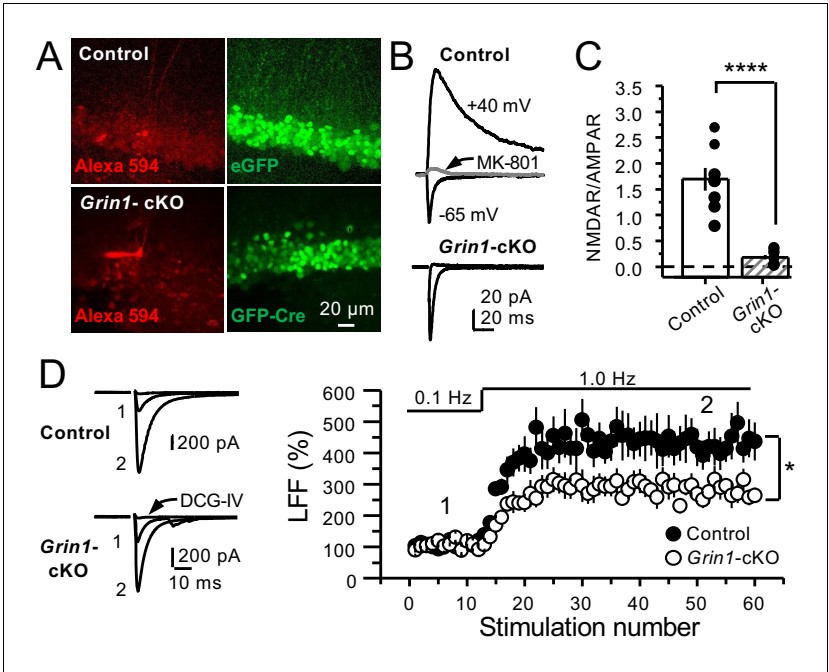

**Figure 2.** GluN1 deletion from GCs reduces mf-CA3 facilitation. (**A**) Representative images showing GCs patch-loaded with Alexa 594 (35 µM) (*left*), and GFP expression in GCs (*right*). (**B**) Representative EPSCs recorded from control (GFP+) and *Grin1*-cKO (Cre-GFP+) GCs. Synaptic responses were elicited by activating medial perforant-path inputs. AMPAR-ESPCs were recorded at $V_h$ = −65 mV in the presence of 100 µM picrotoxin, NMDAR-EPSCs were isolated with 10 µM NBQX and recorded at +40 mV. MK-801 (20 µM) was applied at the end of each recording. (**C**) Summary plot demonstrating that GluN1 deletion from GCs virtually abolished NMDAR-mediated transmission indicated by a strong reduction of NMDAR/AMPAR in *Grin1*-cKO GCs as compared to controls (control 1.61 ± 0.18, n = 9 cells, nine animals, *Grin1*-cKO 0.18 ± 0.04, n = 10 cells, 10 animals; control vs *Grin1*-cKO, p=9.2×10^−6, unpaired t-test). (**D**) LFF was significantly reduced in GluN1-deficient animals (control, 430 ± 5%, n = 13 cells, 10 animals; *Grin1*-cKO, 291 ± 6%, n = 11 cells, 10 animals; p=0.0239, unpaired t-test). Representative traces (*left*) and summary plot (*right*). LFF was induced by stepping stimulation frequency from 0.1 to 1 Hz. DCG-IV (1 µM) was added at the end of each experiment. Data are presented as mean ± s.e.m. *p<0.05; ****p<0.001.

## PreNMDARs boost information transfer by enhancing burst-induced facilitation at mossy fiber synapses

GCs in vivo typically fire in brief bursts (*Diamantaki et al., 2016*; *GoodSmith et al., 2017*; *Henze et al., 2002*; *Pernía-Andrade and Jonas, 2014*; *Senzai and Buzsáki, 2017*). To test whether preNMDARs contribute to synaptic facilitation that occurs during more physiological patterns of activity, mfs were activated with brief bursts (five stimuli, 25 Hz). We first took an optogenetic approach and used a Cre-dependent ChIEF virus to selectively light-activate mf-CA3 synapses in *Grin1*-cKO and control mice. Thus, animals were injected with a mix of AAV5-CaMKII-CreGFP+AAV DJ-FLEX-ChIEF-tdTomato viruses in the DG (*Figure 4A*). At least 4 weeks after surgery, acute slices were prepared and burst-induced facilitation of AMPAR-mediated transmission in CA3 was assessed (*Figure 4B,C*). Burst-induced facilitation, triggered by light stimulation and measured as the ratio of EPSCs elicited by the fifth and first pulse (P5/P1 ratio), was significantly reduced in *Grin1*-cKO animals as compared to controls. Because these bursts of activity can activate the CA3 network (*Henze et al., 2000*; *Kwon and Castillo, 2008*; *Nicoll and Schmitz, 2005*), we next monitored KAR-EPSCs under conditions of low excitability (as in *Figure 1G*). MK-801 bath application also reduced burst-induced facilitation, whereas facilitation remained unchanged in naïve slices (*Figure 4D,E*). In a separate set of experiments, we confirmed the reduction of MK-801 on burst-induced facilitation under more physiological recording conditions (*Figure 4—figure supplement 1*). Lastly, we tested whether preNMDARs, by facilitating glutamate release during bursting activity, could bring CA3 pyramidal neurons to threshold and trigger postsynaptic action potentials. To test this possibility, we monitored action potentials elicited by KAR-EPSPs (resting membrane potential −70 ± 2 mV)

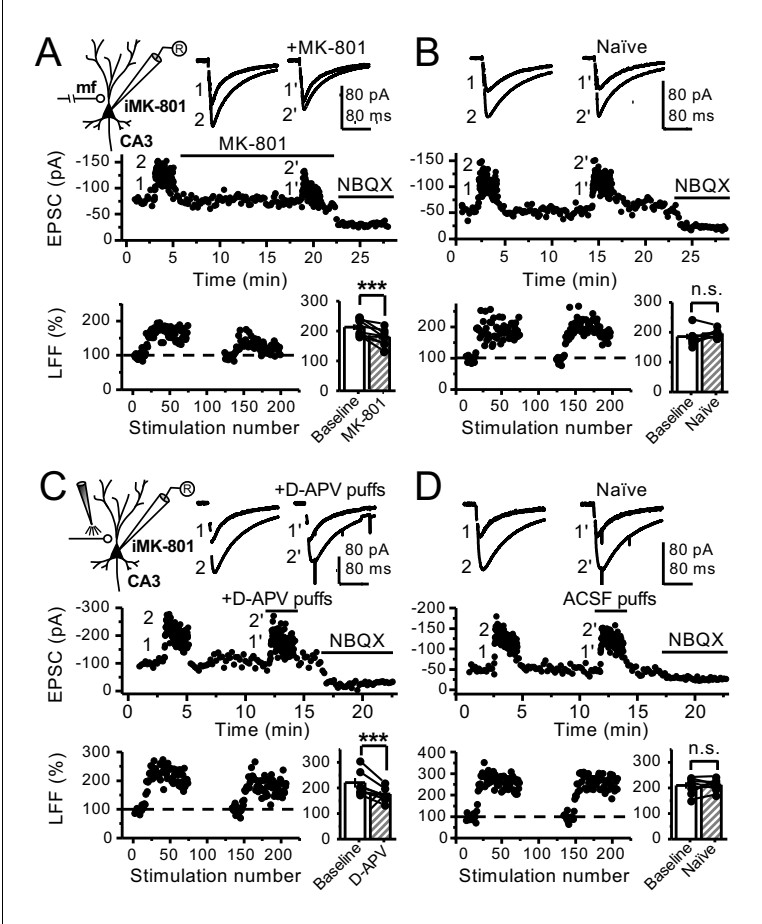

**Figure 3.** Reduced facilitation by NMDAR antagonism is independent of the GC somatodendritic compartment. (A) KAR-EPSCs were recorded at $V_h = -70$ mV in the presence of 15 µM LY303070 and 100 µM picrotoxin. In addition, NMDAR-mediated transmission was blocked intracellularly by loading MK-801 (2 mM) in the patch-pipette. LFF of KAR-EPSCs was assessed as in *Figure 1G* but with transected mf axons (see Materials and methods). Bath application of MK-801 (50 µM) significantly reduced LFF (baseline 213 ± 9%, MK-801 181 ± 10%, n = 8 cells, seven animals; baseline vs MK-801, p=0.002, paired t-test). In all panels of this figure: recording arrangement (*inset*), representative traces (*top*), representative experiment (*middle*), normalized LFF and summary plot (*bottom*). (B) Stable LFF in transected, naïve slices (baseline 186 ± 10%, naïve 196 ± 5%, n = 8 cells, seven animals; baseline vs naïve, p=0.278, paired t-test). (C) LFF was induced before and during puff application of D-APV (2 mM) in *stratum lucidum*. This manipulation significantly reduced facilitation (baseline 220 ± 19%, D-APV puff 176 ± 11%, n = 7 cells, seven animals; baseline vs D-APV puff, p=0.003, paired t-test). (D) Stable LFF in acute slices during puff application of ACSF (baseline 210% ± 12, naïve 213% ± 9, n = 7 cells, seven animals; baseline vs naïve, p=0.778, paired t-test). NBQX (10 µM) was applied at the end of all recordings to confirm mf KAR transmission. Data are presented as mean ± s.e.m. ***p<0.005.

The online version of this article includes the following figure supplement(s) for figure 3:

**Figure supplement 1.** Targeting preNMDARs located in mf axons, but not granule cells.

---

from CA3 pyramidal neurons intracellularly loaded with 2 mM MK-801. Under these recording conditions, MK-801 bath application significantly reduced the mean number of spikes per burst (*Figure 4F*). No changes in mean spikes per burst were observed in naïve slices over time (*Figure 4G*). Application of 10 µM NBQX at the end of these experiments confirmed that action potentials were induced by KAR-mediated synaptic responses. Consistent with these observations, MK-801 also reduced the mean number of spikes per burst when AMPAR-mediated action potentials were recorded from CA3 pyramidal neurons (*Figure 4—figure supplement 2*). In control experiments, we found that intracellular MK-801 effectively blocked postsynaptic NMDAR transmission

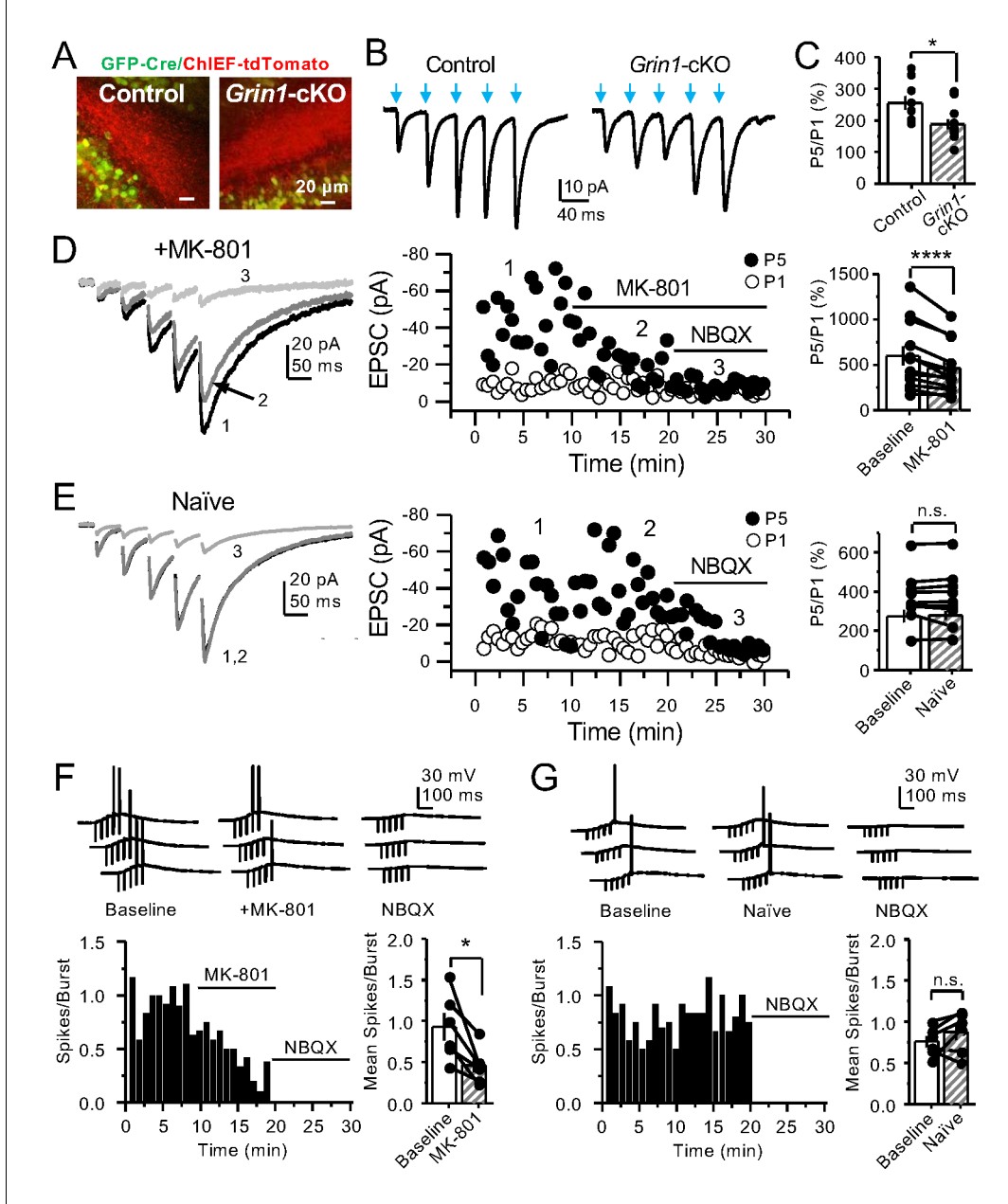

**Figure 4.** PreNMDARs contribute significantly to burst-induced facilitation and spike transfer. (**A**) Representative images showing expression of GFP-Cre (*left*) and ChIEF-tdTomato (*right*) in the DG of control and *Grin1*-cKO animals. (**B**) Representative AMPAR-EPSCs from control (*left*) and *Grin1*-cKO (*right*) CA3 pyramidal neurons recorded at $V_h = -65$ mV and evoked by optical burst-stimulation (5 pulses at 25 Hz) of *stratum lucidum*. Blue arrows indicate light stimulation. (**C**) Summary plot of burst-induced facilitation measured as P5/P1 ratio of optical responses; facilitation was significantly reduced in *Grin1*-cKO animals as compared to control (*Grin1*-cKO 187 ± 16%, n = 12 cells, nine animals; control 255 ± 22%, n = 9 cells, eight animals; *Grin1*-cKO vs control, p=0.0167, unpaired t-test). (**D**) Burst stimulation induced KAR-EPSCs were isolated and recorded as described in *Figure 3*, bath application of MK-801 (50 μM) significantly reduced facilitation (baseline 601 ± 107%, MK-801 464 ± 84%, n = 13 cells, 10 animals; baseline vs MK-801, p=0.00042, paired t-test). In (**D**) and (**E**) of this figure: representative traces (*left*), representative experiment (*middle*), and summary plot (*right*). (**E**) Burst-induced facilitation was stable in interleaved, naïve slices (baseline 369 ± 45%, naïve 367 ± 48%, n = 9 cells, nine animals; p=0.863, paired t-test). (**F**) Bath application of MK-801 (50 μM) reduced KAR-mediated action potentials induced by burst-stimulation (baseline 0.93 ± 0.17, MK-801 0.46 ± 0.09, n = 6 cells, five animals; p=0.036, Wilcoxon signed-rank test). In (**F**) and ( **G**) of this figure: representative traces (*top*), representative experiment and summary plot (*bottom*). (**G**) Stable

*Figure 4 continued on next page*

*Figure 4 continued*

KAR-mediated action potentials in interleaved naïve slices (baseline $0.76 \pm 0.07$, naïve $0.88 \pm 0.1$, n = 6 cells, five animals; p=0.2084, Wilcoxon signed-rank test). NBQX (10 µM) was applied at the end of all experiments in (**D–G**). Data are presented as mean ± s.e.m. *p<0.05; ****p<0.001.

The online version of this article includes the following figure supplement(s) for figure 4:

**Figure supplement 1.** PreNMDARs contribute to burst-induced facilitation in more physiological conditions: 1.2 mM $Mg^{+2}$, 1.2 mM $Ca^{+2}$ and 35°C.

**Figure supplement 2.** PreNMDARs contribute to action potential firing elicited by AMPAR-mediated transmission.

**Figure supplement 3.** Intracellular MK-801 effectively blocked postsynaptic NMDARs elicited by burst stimulation (5 pulses at 25 Hz).

during burst stimulation (*Figure 4—figure supplement 3*). Altogether, these results indicate that preNMDARs at mf-CA3 synapses can contribute to information transfer from the DG to CA3.

## PreNMDARs contribute to presynaptic calcium rise and can be activated by glutamate

PreNMDARs could facilitate glutamate and BDNF release by increasing presynaptic $Ca^{2+}$ rise (*Bouvier et al., 2016*; *Buchanan et al., 2012*; *Corlew et al., 2008*; *Park et al., 2014*). To test this possibility at mf-CA3 synapses, we combined a conditional knockout strategy with $Ca^{2+}$ imaging using two-photon laser scanning microscopy. We first deleted preNMDARs by injecting AAV5-CaM-KII-mCherry-Cre virus in the DG of *Grin1* floxed mice, and littermate animals injected with AAV5-CaMKII-mCherry virus served as control (*Figure 5A*). Two weeks after surgery, we confirmed the efficacy of *Grin1* deletion by activating medial perforant-path inputs and monitoring NMDAR/AMPAR ratios in GCs of control and *Grin1*-cKO animals (*Figure 5A*). Virtually no NMDAR-EPSCs were detected at $V_h$ = +40 mV in *Grin1*-cKO animals (*Figure 5A*). Acute slices that exhibited robust mCherry fluorescence in the DG were used for $Ca^{2+}$ imaging experiments. To maximize our ability to detect preNMDAR-mediated $Ca^{2+}$ signals, we used a recording solution that contained 0 mM $Mg^{2+}$, 4 mM $Ca^{2+}$, and 10 µM D-Serine (*Carter and Jahr, 2016*). GCs expressing mCherry were patch-loaded with 35 µM Alexa 594 (used as morphological dye) and 200 µM Fluo-5F, and mf axons were imaged and followed toward CA3 until giant boutons (white arrows) were identified (*Figure 5B*). We found that $Ca^{2+}$ transients (CaTs) elicited by direct current injection in the GC soma (five action potentials, 25 Hz) were significantly smaller in *Grin1*-cKO animals as compared to control (*Figure 5C–E*). In addition, NMDAR antagonism with D-APV reduced presynaptic $Ca^{2+}$ rise even under more physiological $Mg^{+2}$ concentration in acute rat hippocampal slices (*Figure 5—figure supplement 1*). Thus, preNMDARs contribute significantly to presynaptic $Ca^{2+}$ rise in mf boutons, and by this means likely facilitates synaptic transmission, although a potential contribution of $Ca^{2+}$ rise-independent effects cannot be discarded.

Lastly, we sought to determine whether direct activation of preNMDARs could drive $Ca^{2+}$ influx in mf giant boutons. To test this possibility, we elicited CaTs by two-photon glutamate uncaging (2PU) on mf boutons of control and *Grin1*-cKO animals (*Figure 6A*). As previously described, mCherry GCs were patch-loaded with Alexa 594 and Fluo-5F in a recording solution designed to maximize the detection of preNMDAR-mediated $Ca^{2+}$ signals (as in *Figure 5*). We first confirmed that glutamate 2PU-induced CaTs in dendritic spine heads of GCs were strongly reduced in *Grin1*-cKO animals as compared to controls (*Figure 6B,C*). To verify that reduced $Ca^{2+}$ signals ($\Delta G/R$) were a result of *Grin1* deletion and not differences in uncaging laser power, we performed a laser power intensity–response curve and found that *Grin1*-cKO animals exhibited reduced $\Delta G/R$ signals as compared to control regardless of laser power intensity (*Figure 6—figure supplement 1*). We next measured glutamate 2PU-induced CaTs in mf giant boutons (identified as in *Figure 5B*) and found that single uncaging pulses were insufficient to drive detectable CaTs in control boutons (*Figure 6—figure supplement 2*). However, a burst of 2PU stimulation (5 pulses, 25 Hz) induced CaTs in mf boutons of control but not in *Grin1*-cKO animals (*Figure 6D,E*). Additionally, CaTs elicited by 2PU stimulation were abolished by D-APV application (*Figure 6—figure supplement 3*). These findings indicate that brief bursts of glutamate 2PU, a manipulation that mimics endogenous release of glutamate during physiological patterns of activity, induces presynaptic $Ca^{2+}$ influx in mf boutons by activating preNMDARs.

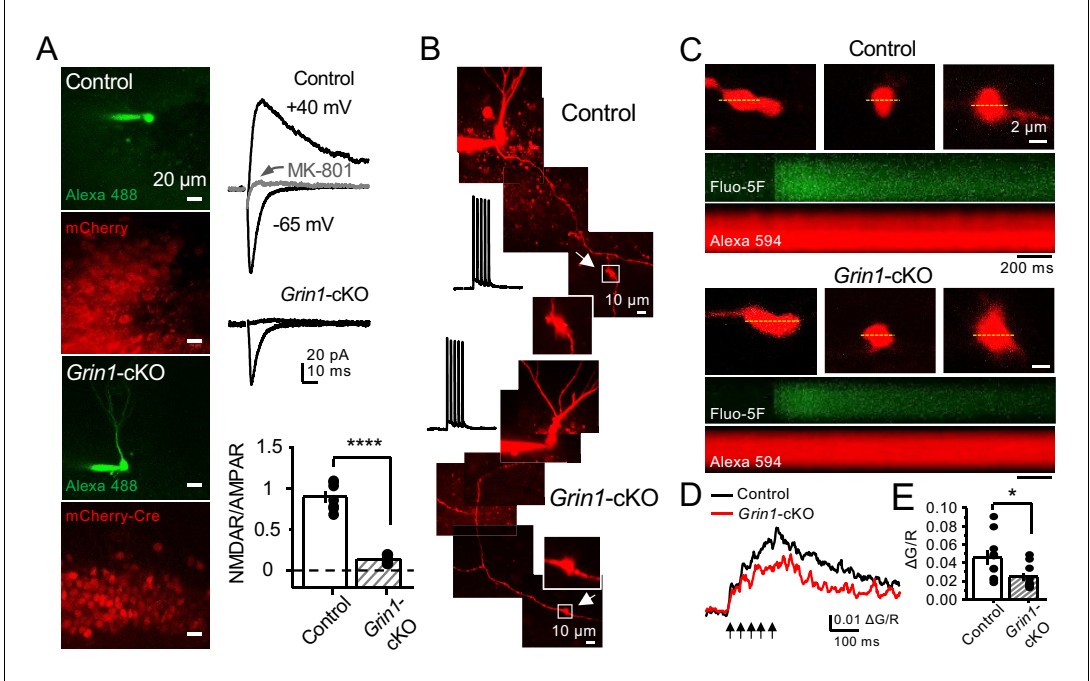

**Figure 5.** preNMDARs contribute to presynaptic $Ca^{2+}$ rise. (**A**) Representative images showing GCs patch-loaded with Alexa 488 (35 µM) to confirm expression of mCherry (*bottom*). Representative AMPAR-EPSCs recorded from control (*top*) or *Grin1*-cKO (*middle*) GCs. Synaptic responses were elicited by activating medial perforant-path inputs. AMPAR-ESPCs were recorded at $V_h = -65$ mV in the presence of 100 µM picrotoxin, NMDAR-EPSCs were isolated with 10 µM NBQX and recorded at +40 mV. MK-801 (20 µM) was applied at the end of each experiment. Summary plot (*bottom*) demonstrating that GluN1 deletion from GCs virtually abolished NMDAR-mediated transmission indicated by a strong reduction of NMDAR/AMPAR in *Grin1*-cKO granule cells as compared to controls (control 0.90 ± 0.17, n = 7 cells, six animals; *Grin1*-cKO 0.13 ± 0.05, n = 6 cells, six animals; control vs *Grin1*-cKO, p=3.81×$10^{-7}$, unpaired t-test). (**B**) Representative control and *Grin1*-cKO GCs patch-loaded with Fluo-5F (200 µM) and Alexa 594 (35 µM). Arrows indicate the identification of a mf giant bouton, magnified images in white box. (**C**) Three representative mf boutons (*top*) and line scan image of calcium transients (CaTs) elicited by five action potentials at 25 Hz (middle, Fluo-5F) and morphological dye (*bottom*, Alexa 594), in Control and *Grin1*-cKO animals. Dotted line (yellow) indicates line scan location. Red Channel, Alexa 594; Green Channel, Fluo-5F. (**D, E**) Peak analysis of the fifth pulse ΔG/R revealed a significant reduction in $Ca^{2+}$ rise of *Grin1*-cKO animals as compared to Control (control 0.046 ± 0.01, n = 10 boutons, three line scans per bouton, eight animals; *Grin1*-cKO 0.025 ± 0.004, n = 10 boutons, eight animals; control vs *Grin1*-cKO, U = 0.017, Mann–Whitney test). Arrows indicate mf activation. Data are presented as mean ± s.e.m. *U < 0.05; ****p<0.001.

The online version of this article includes the following figure supplement(s) for figure 5:

**Figure supplement 1.** NMDAR antagonism reveals a reduction in presynaptic $Ca^{+2}$ rise in the presence of 1.3 mM $Mg^{+2}$ and 2.5 mM $Ca^{+2}$.

## PreNMDARs promote BDNF release from mossy fiber boutons

Previous work implicated preNMDARs in the release of BDNF at corticostriatal synapses following burst stimulation and presynaptic $Ca^{2+}$ elevations (*Park et al., 2014*). Given the high expression levels of BDNF in mfs (*Conner et al., 1997*; *Yan et al., 1997*), we examined the potential role for pre-NMDARs in BDNF release from mf terminals. To this end, a Cre-dependent BDNF reporter (BDNF-pHluorin) was injected in *Grin1*-floxed and control animals. Littermate mice were injected with a mix of AAV5-CaMKII-mCherry-Cre + AAV-DJ-DIO-BDNF-pHluorin viruses in the DG (*Figure 7A*). At least 4 weeks after surgery, acute slices were prepared for two-photon laser microscopy to image mf boutons. After acquiring a stable baseline of BDNF-pHluorin signals, mfs were repetitively activated (see Materials and methods) (*Figure 7B*). BDNF-pHluorin signals were analyzed by measuring ΔF/F, where ΔF/F reductions indicate BDNF release (*Park et al., 2014*). We found that GluN1-deficient mf boutons showed a significant (~50%) impairment in BDNF release as compared to control (*Figure 7C–D*). Furthermore, using a more physiological pattern of burst stimulation, GluN1-lacking mf boutons still displayed altered BDNF release as compared to control (*Figure 7—figure supplement 1*). Taken together, our results suggest preNMDARs contribute significantly to BDNF release during repetitive or burst stimulation of mf synapses.

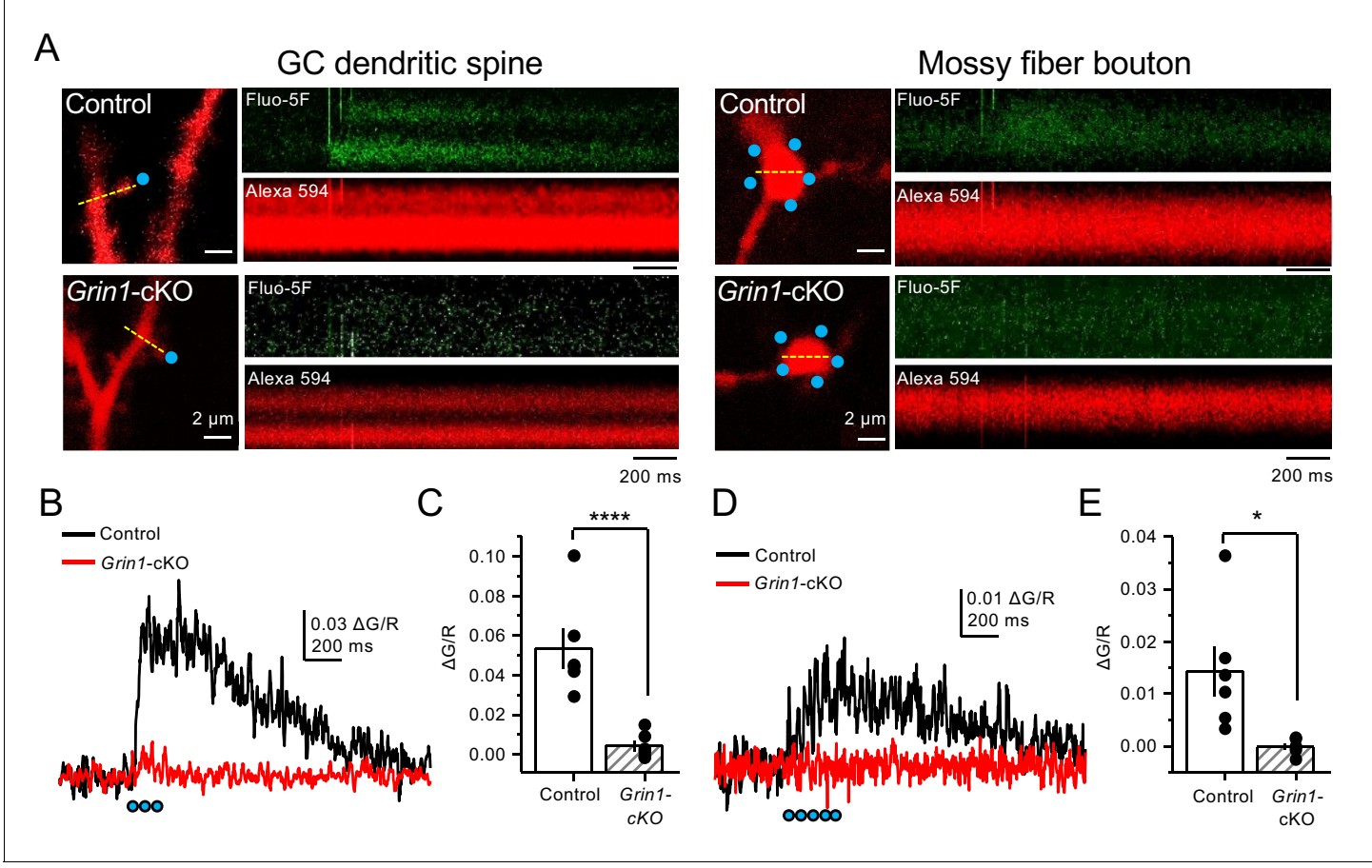

**Figure 6.** Uncaging glutamate induces Ca$^{2+}$ rise in mossy fiber boutons. (**A**) Representative images showing dendritic spines in GCs (*left*) and mf boutons (*right*), and the associated line scan image of calcium transients (CaTs) elicited by uncaging of MNI-glutamate (see Materials and methods), in control and *Grin1*-cKO animals. Blue dots indicate uncaging spots. Red channel, Alexa 594; Green channel, Fluo-5F. (**B**) Line scan analysis of CaTs measuring ΔG/R in dendritic spines when MNI-glutamate is uncaged in control or *Grin1*-cKO animals. Blue dots indicate location of two-photon uncaging (2PU) pulses. (**C**) Summary plot demonstrating a significant reduction in dendritic spine CaTs in *Grin1*-cKO as compared to Control (control 0.053 ± 0.01 ΔG/R, n = 6 dendritic spines, three line scans per spine, six animals; *Grin1*-cKO 0.004 ± 0.003 ΔG/R, n = 6 spines, three line scans per spine, six animals; ΔG/R control vs *Grin1*-cKO, p=0.00088, unpaired t-test). (**D**) Line scan analysis of CaTs measuring ΔG/R in mf boutons when MNI-glutamate is uncaged in control or *Grin1*-cKO animals. (**E**) Summary plot demonstrating significant CaTs in boutons of control as compared to *Grin1*-cKO (control 0.014 ± 0.005, n = 6 boutons, three line scans per bouton, six animals; *Grin1*-cKO −0.00012 ± −0.0006, n = 6 boutons, three line scans per bouton, six animals; control vs *Grin1*-cKO, p=0.015, unpaired t-test). Data are presented as mean ± s.e.m. *p<0.05; ****p<0.001.

The online version of this article includes the following figure supplement(s) for figure 6:

**Figure supplement 1.** *Grin1*-cKO exhibit reduced CaTs at varying uncaging laser power intensities.

**Figure supplement 2.** Bouton CaTs can be detected after repetitive uncaging of MNI-glutamate.

**Figure supplement 3.** NMDAR antagonism with D-APV blocks CaTs elicited by glutamate 2PU.

## PreNMDAR-mediated regulation of mossy fiber synapses is input specific

In addition to providing a major excitatory input to the hippocampus proper, mf axons also synapse onto excitatory hilar MCs and inhibitory neurons in CA3 (*Amaral et al., 2007*; *Henze et al., 2000*; *Lawrence and McBain, 2003*). To test whether preNMDARs could also play a role at these synapses, we visually patched MCs and INs in acute rat hippocampal slices, loaded them with 35 µM Alexa 594 (*Figure 8A*) and 2 mM MK-801, and monitored AMPAR-EPSCs (V$_h$ = −70 mV). Unlike mf-CA3 synapses, mf synapses onto CA3 INs in *stratum lucidum* do not express LFF, but can undergo burst-induced facilitation or depression (*Toth et al., 2000*). We found that MK-801 bath application had no effect on burst-induced facilitation or depression (*Figure 8B*), suggesting preNMDARs do not play a role at mf-IN synapses in CA3. Mf inputs onto hilar MCs undergo robust activity-dependent

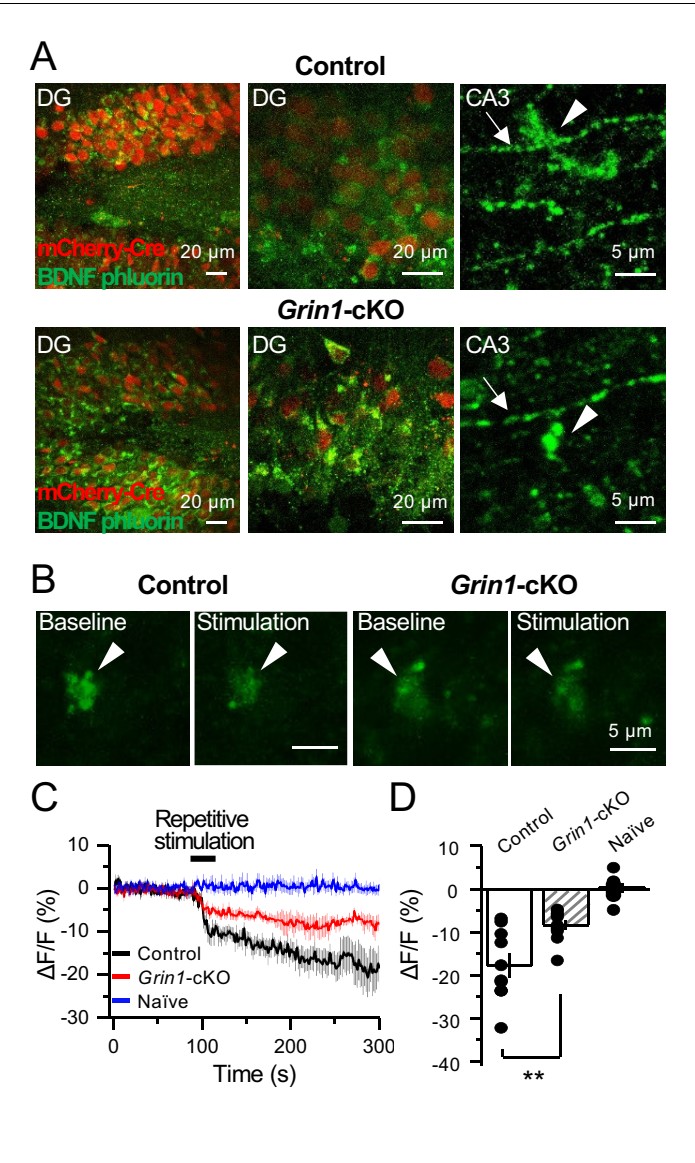

**Figure 7.** preNMDARs contribute significantly to BDNF release following repetitive activity. (**A**) Representative images showing expression of BDNF-pHluorin in the DG and CA3 area (arrows indicate mf axon, arrowheads indicate mf boutons). Control images (*top*), *Grin1*-cKO images (*bottom*). (**B**) Representative images of BDNF-pHluorin signal intensity at baseline and after repetitive stimulation of mfs (125 pulses, 25 Hz, ×2). Control images (*left*), *Grin1*-cKO images (*right*), arrowhead indicates region of interest. (**C**) Time course of BDNF-pHluorin signal intensity measured as ΔF/F (%): control (*black*), *Grin1*-cKO (red), Naïve (blue). (**D**) Quantification of BDNF-pHluorin signal in (**C**) during the last 100 s reveals larger BDNF release in control animals as compared to *Grin1*-cKO (control −18% ± 3%, n = 9 slices, five animals; *Grin1*-cKO −8% ± 1%, n = 10 slices, five animals; *Grin1*-cKO vs control, p=0.00648, unpaired t-test). Data are presented as mean ± s.e.m. **p<0.01.

The online version of this article includes the following figure supplement(s) for figure 7:

**Figure supplement 1.** preNMDARs contribute significantly to BDNF release following a more physiological pattern of burst stimulation.

facilitation (*Lysetskiy et al., 2005*). Similar to mf-CA3 synapses, we found that MK-801 reduced LFF (*Figure 8C*). Stability experiments of mf transmission at CA3 INs or hilar MCs showed no significant differences (*Figure 8—figure supplement 1*). Taken together, our findings demonstrate that pre-NMDARs facilitate mf transmission onto excitatory neurons, but not onto inhibitory INs.

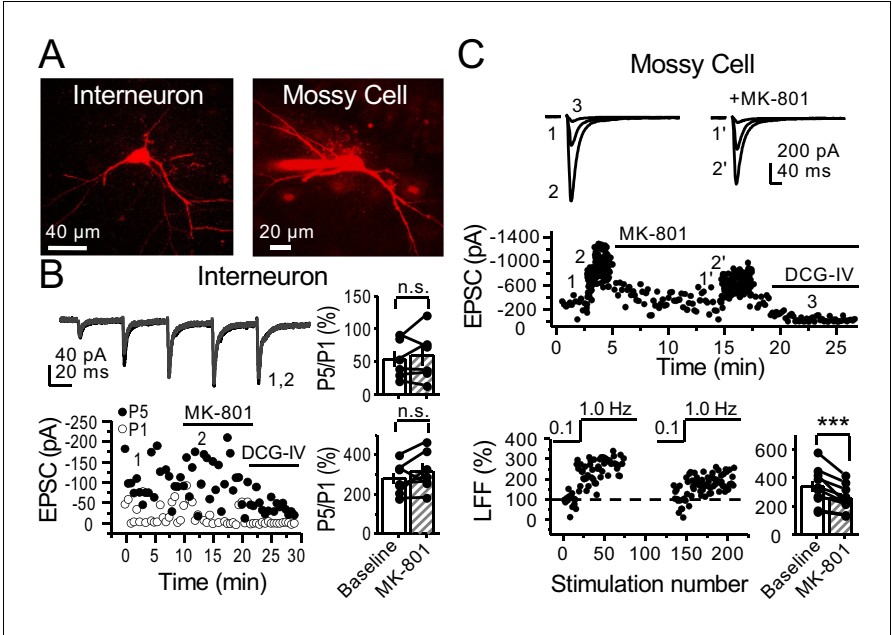

**Figure 8.** preNMDARs contribute to synaptic facilitation of mossy fiber inputs onto mossy cells, but not onto CA3 inhibitory interneurons. (**A**) Representative images showing a CA3 IN and a hilar MC patch-loaded with Alexa 594 (35 µM) for morphological identification in acute rat hippocampal slices. (**B**) AMPAR-EPSCs were recorded from CA3 INs at $V_h = -65$ mV and burst stimulation was elicited by 5 pulses at 25 Hz, see traces (*top*). Representative experiment (*bottom, left*), and summary plots (*right*) showing bath application of MK-801 (50 µM) had no significant effect on depression (*top, right*) or facilitation (*bottom, right*) measured by P5/P1 ratio (baseline 54 ± 12%, MK-801 60 ± 16%, n = 6 cells; MK-801 vs baseline, p=0.675, Wilcoxon signed-rank test; baseline 281 ± 30%, MK-801 318 ± 37%, n = 7 cells; MK-801 vs baseline, p=0.178, paired t-test, five animals in each data set). (**C**) AMPAR-ESPCs were recorded at $V_h = -70$ mV from MCs, LFF was induced by stepping stimulation frequency from 0.1 to 1 Hz, see traces (*top*). Representative experiment (*middle*), normalized LFF and summary plot (*bottom*) indicating bath application of MK-801 (50 µM) reduced facilitation (baseline 339 ± 41%, MK-801 258 ± 29%, n = 10 cells, six animals; baseline vs MK-801, p=0.00152, paired t-test). DCG-IV (1 µM) was applied at the end of all experiments. Data are presented as mean ± s.e.m. ***p<0.005.
The online version of this article includes the following figure supplement(s) for figure 8:

**Figure supplement 1.** Stability experiments for mf-Interneuron and mf-mossy cell short-term plasticity.

## Discussion

In this study, we provide evidence that hippocampal mf boutons express preNMDARs whose activation fine-tunes mf synaptic function. Specifically, our results show that preNMDARs enhance mf short-term plasticity in a target cell-specific manner. By enhancing glutamate release onto excitatory neurons but not inhibitory INs, preNMDARs increase GC-CA3 spike transfer. Moreover, using two-photon $Ca^{2+}$ imaging, we demonstrate that preNMDARs contribute to presynaptic $Ca^{2+}$ rise in mf boutons. Lastly, upon repetitive activity, preNMDARs promote BDNF release from mf boutons. Taken together, our findings indicate that preNMDARs act as autoreceptors to boost both glutamate and BDNF release at mf synapses. By regulating information flow in the DG-CA3 circuit, preNMDARs may play a significant role in learning and memory.

Early studies using immunoperoxidase electron microscopy revealed NMDARs in presynaptic compartments in multiple brain areas (for a review, see *Corlew et al., 2008*). Subsequent studies that used immunogold electron microscopy, a more precise localization method, identified NMDARs on the presynaptic membrane in a number of brain structures, including neocortex (*Fujisawa and Aoki, 2003*; *Larsen et al., 2011*), hippocampus (*Berg et al., 2013*; *Jourdain et al., 2007*; *McGuinness et al., 2010*), and amygdala (*Pickel et al., 2006*). In agreement with these studies, and using a previously validated antibody (*Siegel et al., 1994*), we identified prominent presynaptic labeling of the obligatory subunit GluN1 in mf boutons (*Figure 1A–D*). Moreover, we found that

these receptors are close to the active zone and therefore well positioned to modulate neurotransmitter release.

Previous work in the cerebellum and neocortex suggested that somatodendritic potentials generated by NMDARs could signal to nerve terminals and lead to presynaptic $Ca^{2+}$ elevations (*Christie and Jahr, 2008*; *Christie and Jahr, 2009*). Thus, changes in neurotransmitter release resulting from NMDAR antagonism could be due to somatodendritic NMDARs but not necessarily preNMDARs residing on nerve terminals (*Duguid, 2013*). However, we showed that focal NMDAR antagonism far from the somatodendritic compartment and in transected axons still reduced short-term plasticity at mf synapses (*Figure 3*), making it extremely unlikely that somatodendritic NMDARs could explain our results. In further support of functional preNMDARs at mf boutons, we found that 2PU of glutamate induced $Ca^{2+}$ rise in control, but not in GluN1-deficient boutons. Together, our findings strongly support the presence of functional preNMDARs facilitating neurotransmission at mf-CA3 synapses.

There is evidence that preNMDARs can operate as coincidence detectors at some synapses (*Duguid, 2013*; *Wong et al., 2021*). At the mf-CA3 synapse, we found that preNMDARs contribute to LFF (i.e. 1 s inter-stimulus interval). This observation is intriguing given that the presynaptic AP-mediated depolarization is likely absent by the time glutamate binds to preNMDARs. However, coincidence detection may not be an essential requirement for mf preNMDARs to modulate glutamate release. Of note, at resting membrane potential, the NMDAR conductance is not zero and the driving force for $Ca^{2+}$ influx is high (*Paoletti et al., 2013*; *Traynelis et al., 2010*). It is also conceivable that mf preNMDARs exhibit low-voltage dependence, as it has been reported at other synapses (*Wong et al., 2021*). Remarkably, the somatodendritic compartment of GCs can generate sub-threshold depolarizations at mf terminals (a.k.a. excitatory presynaptic potentials) (*Alle and Geiger, 2006*). By alleviating the magnesium blockade, these potentials might reduce the need for coincidence detection and transiently boost the functional impact of mf preNMDARs.

While the presence of preNMDARs is downregulated during development both in neocortex (*Corlew et al., 2007*; *Larsen et al., 2011*) and in hippocampus (*Mameli et al., 2005*), we were able to detect functional preNMDARs in young adult rats (P17–P28) and mice (P30–P44), once mf connections are fully developed (*Amaral and Dent, 1981*). Functional preNMDARs have been identified in axonal growth cones of hippocampal and neocortical neurons, suggesting that these receptors are important for regulating early synapse formation (*Gill et al., 2015*; *Wang et al., 2011*). Because GCs undergo adult neurogenesis, and adult-born GCs establish new connections in the mature brain, preNMDARs could also play an important role at immature mf synapses and functional integration of new born GCs into the mature hippocampus (*Toni and Schinder, 2015*). Moreover, experience can modulate the expression and composition of preNMDARs in neocortex (*Larsen et al., 2014*), a possibility not investigated in our study.

The glutamate that activates preNMDARs may originate from the presynaptic terminal, the postsynaptic cell, nearby synapses or neighboring glial cells. Our results indicate that activation of preNMDARs at mf synapses requires activity-dependent release of glutamate that likely arises from mf boutons, although other sources cannot be discarded, including astrocytes. For instance, at medial entorhinal inputs to GCs, preNMDARs appear to be localized away from the presynaptic release sites and facing astrocytes, consistent with preNMDAR activation by gliotransmitters (*Jourdain et al., 2007*; *Savtchouk et al., 2019*). In contrast, at mf-CA3 synapses, we found that preNMDARs are adjacent to the release sites, suggesting a direct control on glutamate release from mf boutons.

The precise mechanism by which preNMDARs facilitate neurotransmitter release is poorly understood, but it may include $Ca^{2+}$ influx through the receptor and depolarization of the presynaptic terminal with subsequent activation of voltage-gated $Ca^{2+}$ channels (*Banerjee et al., 2016*; *Corlew et al., 2008*). In support of this mechanism is the high $Ca^{2+}$ permeability of NMDARs (*Paoletti et al., 2013*; *Rogers and Dani, 1995*). Besides, presynaptic sub-threshold depolarization and subsequent activation of presynaptic voltage-gated $Ca^{2+}$ channels is a common mechanism by which presynaptic ionotropic receptors facilitate neurotransmitter release (*Engelman and MacDermott, 2004*; *Pinheiro and Mulle, 2008*). PreNMDARs may also act in a metabotropic manner (*Dore et al., 2016*) and facilitate spontaneous transmitter release independent of $Ca^{2+}$ influx (*Abrahamsson et al., 2017*). Our findings demonstrating that the open channel blocker MK-801 robustly reduced short-term plasticity at mf synapses support an ionotropic mechanism that involves

Ca$^{2+}$ influx through preNMDARs. A previous study failed to observe Ca$^{2+}$ reductions in mf boutons by DL-APV (*Liang et al., 2002*). A combination of factors could account for this discrepancy with our study, including a stronger mf repetitive stimulation (20 pulses, 100 Hz), which may overcome a less potent NMDAR antagonism and/or the need for preNMDAR activity, as well as the use of a higher affinity Ca$^{2+}$ indicator (Fura-2 AM) and a lower spatiotemporal resolution imaging approach. Nevertheless, in line with previous studies that detected presynaptic Ca$^{2+}$ rises following local activation of NMDARs (e.g. NMDA or glutamate uncaging) in visual cortex (*Buchanan et al., 2012*) and cerebellum (*Rossi et al., 2012*), we provide direct evidence that preNMDAR activation by either repetitive activation of mfs or 2PU of glutamate increases presynaptic Ca$^{2+}$ (*Figures 5* and *6*). Although the Ca$^{2+}$ targets remain unidentified, these may include proteins of the release machinery, calcium-dependent protein kinases and phosphatases, and Ca$^{2+}$ release from internal stores (*Banerjee et al., 2016*). In addition to facilitating evoked neurotransmitter release, preNMDARs can promote spontaneous neurotransmitter release as indicated by changes in miniature, action potential-independent activity (e.g. mEPSCs) (for recent reviews, see *Banerjee et al., 2016*; *Kunz et al., 2013*; *Wong et al., 2021*). A potential role for preNMDARs in spontaneous, action potential-independent release at mf synapses cannot be discarded.

Our results show that activation of preNMDARs by physiologically relevant patterns of presynaptic activity enhanced mf transmission and DG-CA3 information transfer (*Figure 4*). A previous study reported that NMDAR genetic deletion in GCs resulted in memory deficits (e.g. pattern separation) (*McHugh et al., 2007*). Although the mechanism is unclear, it could involve activity-dependent preNMDAR regulation of mf excitatory connections. We also found that preNMDARs facilitate neurotransmitter release in a target cell-specific manner. Like in neocortex (*Larsen and Sjöström, 2015*), such specificity strongly suggests that preNMDARs have distinct roles in controlling information flow in cortical microcircuits. Thus, preNMDAR facilitation of mf synapses onto glutamatergic neurons but not GABAergic INs (*Figure 8*) may fine-tune the CA3 circuit by increasing the excitatory/inhibitory balance.

Given the multiple signaling cascades known to regulate NMDARs (*Lau and Zukin, 2007*; *Sanz-Clemente et al., 2013*), preNMDARs at mf synapses may provide an important site of neuromodulatory control. PreNMDARs have been implicated in the induction of LTP and LTD at excitatory or inhibitory synapses in several brain areas (*Banerjee et al., 2016*; *Wong et al., 2021*). While most evidence, at least using robust induction protocols in vitro, indicates that long-term forms of presynaptic plasticity at mf synapses can occur in the absence of NMDAR activation (*Castillo, 2012*; *Nicoll and Schmitz, 2005*), our findings do not discard the possibility that preNMDARs could play a role in vivo during subtle presynaptic activities. As previously reported for corticostriatal LTP (*Park et al., 2014*), preNMDARs could regulate long-term synaptic plasticity by controlling BDNF release, which is consistent with BDNF-TrkB signaling being implicated in mf-CA3 LTP (*Schildt et al., 2013*). In addition, BDNF could facilitate glutamate release by enhancing NMDAR function at the presynapse, as previously suggested (*Chen et al., 2014*; *Madara and Levine, 2008*), although the precise mechanism(s) remain unclear. By potentiating mf-CA3 transmission, BDNF could also promote epileptic activity (*McNamara and Scharfman, 2012*). Lastly, dysregulation of NMDARs is commonly implicated in the pathophysiology of brain disorders such as schizophrenia, autism, and epilepsy (*Lau and Zukin, 2007*; *Paoletti et al., 2013*). PreNMDAR expression and function have been suggested to be altered in experimental models of disease, including neuropathic pain (*Chen et al., 2019*; *Zeng et al., 2006*) and epilepsy (*Yang et al., 2006*). At present, however, in vivo evidence for the involvement of preNMDARs in brain function and disease is rather indirect (*Bouvier et al., 2015*; *Wong et al., 2021*). The development of specific preNMDAR tools is required to determine the functional impact of these receptors in vivo.

## Materials and methods

### Key resources table

| Reagent type (species) or resource | Designation | Source or reference | Identifiers | Additional information |
|---|---|---|---|---|

*Continued on next page*

*Continued*

| Reagent type (species) or resource | Designation | | Source or reference | Identifiers | Additional information |
|---|---|---|---|---|---|
| Strain, strain background (*Rattus norvegicus* male and female) | Rat: Sprague-Dawley | | Charles River | Strain code: 400 | |
| Strain, strain background (*Mus musculus*, male and female) | Mouse: *Grin1*<sup>fl/fl</sup> (B6.129S4-*Grin1*<sup>tm2Stl</sup>/J) | | Dr. Michael Higley/ The Jackson Laboratory | RRID:IMSR_JAX005246 | |
| Strain, strain background (*Mus musculus* male and female) | Mouse: C57Bl6/J | | Charles River | Strain code: 027 | |
| Antibody | (Include host species and clonality) Mouse, Monoclonal, anti-NMDAR1 | | Millipore | Cat# MAB363 | 10 µg/mL |
| Antibody | Rabbit, Polyclonal, anti-GluA1-4, (pan-AMPA) | | Dr. Elek Molnar/Bristol University | Generated by Dr. Elek Molnar | 10 µg/mL |
| Antibody | Goat anti-rabbit IgG conjugated gold particles | | Nanoprobes Inc | #2003–0.5 ML | (1:100) |
| Recombinant DNA reagent | AAV5-CaMKII-GFP-Cre | | Penn Vector Core | AV-5-PV2521 | Available on Addgene |
| Recombinant DNA reagent | AAV5-CaMKII-eGFP | | Penn Vector Core | AV-5-PV1917 | Available on Addgene |
| Recombinant DNA reagent | AAV5-CaMKII-mcherry-Cre | | UNC Vector Core | See website | https://wwwmed.unc.edu/genetherapy/vectorcore/in-stock-aav-vectors/ |
| Recombinant DNA reagent | AAV5-CaMKII-mcherry | | UNC Vector Core-Dr. Karl Deisseroth Control Fluorophores | See website | https://wwwmed.unc.edu/genetherapy/vectorcore/in-stock-aav-vectors/ |
| Recombinant DNA reagent | AAV-DJ-flex-OChIEF-tdTomato | | Dr. Pascal Kaeser PMID:29398114 | Generated at UNC Vector Core | Custom Order |
| Recombinant DNA reagent | AAV-DJ-DIO-BDNF-phluorin | | Dr. Hyungju Park PMID:25467984 | Generated at UNC Vector Core | Custom Order |
| Chemical compound, drug | Ketamine | | Merial | Cat# 03661103001904 | |
| Chemical compound, drug | Xylazine | | Calier | Cat# 20100–003 | |
| Chemical compound, drug | Paraformaldehyde | | Scharlau | PA0095 | |
| Chemical compound, drug | Glutaraldehyde | | Electron Microscopy Sciences | Cat# 16210 | |
| Chemical compound, drug | Picric Acid | | Panreac | Cat# 141048.1609 | |
| Chemical compound, drug | Phosphate Buffer | | Scharlau | SO03321000 | |
| Chemical compound, drug | Human serum albumin | | SigmaMillipore | A-1653 | |
| Chemical compound, drug | TBS | TRIZMA BASE | SigmaMillipore | T1503 | |
| | | Trizma HCl | | T3253 | |

*Continued on next page*

*Continued*

| Reagent type (species) or resource | Designation | Source or reference | Identifiers | Additional information |
|---|---|---|---|---|
| Chemical compound, drug | Triton X-100 | SigmaMillipore | T8787 | |
| Chemical compound, drug | Polyethylene glycol | SigmaMillipore | 25322-68-3 | |
| Chemical compound, drug | Uranyl acetate | Electron Microscopy Sciences | Cat# 22400 | |
| Chemical compound, drug | Reynold's lead citrate | Electron Microscopy Sciences | #17800 | |
| Chemical compound, drug | Picrotoxin | SigmaMillipore | Cat# P1675 | |
| Chemical compound, drug | LY303070 | ABX Chemical Co. | N/A | Custom Order |
| Chemical compound, drug | MK-801 | Tocris Bioscience | Cat# 0924 | |
| Chemical compound, drug | DCG-IV | Tocris Bioscience | Cat# 0975 | |
| Chemical compound, drug | D-APV | Tocris Bioscience | Cat# 0106 | |
| chemical compound, drug | D-APV | NIMH Chemical Synthesis Program | N/A | |
| Chemical compound, drug | R-CPP | Tocris Bioscience | Cat# 0247 | |
| Chemical compound, drug | NBQX | Cayman Chemical Co. | Cat# 14914 | |
| Chemical compound, drug | Fluo5-F pentapotassium salt cell impermeant | Invitrogen Molecular Probes | Cat# F14221 | |
| Chemical compound, drug | Alexa Fluor 594 Hydrazide | Invitrogen Molecular Probes | Cat# A10438 | |
| Chemical compound, drug | Alexa Fluor 488 Hydrazide | Invitrogen Molecular Probes | Cat# A10436 | |
| Chemical compound, drug | D-Serine | Tocris Bioscience | Cat# 0226 | |
| Chemical compound, drug | MNI-caged-L-glutamate | Tocris Bioscience | Cat# 1490 | |
| Chemical compound, drug | Sucrose | SigmaMillipore | Cat# S9378 | |
| Chemical compound, drug | KCl | SigmaMillipore | Cat# P3911 | |
| Chemical compound, drug | $NaH_2PO_4$ | SigmaMillipore | Cat# S9638 | |
| Chemical compound, drug | $CaCl_2$ | SigmaMillipore | Cat# C8106 | |
| Chemical compound, drug | $MgCl_2$ | SigmaMillipore | Cat# M2670 | |
| Chemical compound, drug | $MgSO_4$ | SigmaMillipore | Cat# M1880 | |
| Chemical compound, drug | Glucose | SigmaMillipore | Cat# G8270 | |
| Chemical compound, drug | NaCl | SigmaMillipore | Cat# S7653 | |
| Chemical compound, drug | $NaHCO_3$ | SigmaMillipore | Cat# S6014 | |

*Continued on next page*

*Continued*

| Reagent type (species) or resource | Designation | Source or reference | Identifiers | Additional information |
|---|---|---|---|---|
| Chemical compound, drug | Cesium hydroxide | SigmaMillipore | Cat# 23204 | |
| Chemical compound, drug | D-gluconic acid | SigmaMillipore | Cat# G1951 | |
| Chemical compound, drug | EGTA | SigmaMillipore | Cat# E4378 | |
| Chemical compound, drug | HEPES | SigmaMillipore | Cat# H3375 | |
| Chemical compound, drug | Potassium gluconate | SigmaMillipore | Cat# G4500 | |
| Chemical compound, drug | MgATP | SigmaMillipore | Cat# A9187 | |
| Chemical compound, drug | $Na_3GTP$ | SigmaMillipore | Cat# G0635 | |
| Chemical compound, drug | NMDG | SigmaMillipore | Cat# M2004 | |
| Chemical compound, drug | Sodium ascorbate | SigmaMillipore | Cat# A4034 | |
| Chemical compound, drug | Thiourea | SigmaMillipore | Cat# T8656 | |
| Chemical compound, drug | Sodium pyruvate | SigmaMillipore | Cat# P2256 | |
| Chemical compound, drug | $KMeSO_4$ | SigmaMillipore | Cat# 83000 | |
| Chemical compound, drug | $Na_2ATP$ | SigmaMillipore | Cat# A2383 | |
| Chemical compound, drug | NaGTP | SigmaMillipore | Cat# 51120 | |
| Chemical compound, drug | Sodium phosphocreatine | SigmaMillipore | Cat# P7936 | |
| Chemical compound, drug | $NH_4Cl$ | SigmaMillipore | Cat# A9434 | |
| Chemical compound, drug | KOH | EMD Millipore | Cat# 109108 | |
| Chemical compound, drug | HCl | Fisher Chemical | Cat# SA49 | |
| Software, algorithm | IgorPro7 | Wavemetrics | | https://www.wavemetrics.com/ |
| Software, algorithm | Origin Pro 9 | Origin Lab | | https://www.originlab.com/ |
| Software, algorithm | ImageJ | ImageJ | | http://imagej.net/Welcome |
| Software, algorithm | Multiclamp 700B | Molecular Devices | | https://www.moleculardevices.com/ |
| Software, algorithm | Prairie View 5.4 | Bruker Corp. | | https://www.pvupdate.blogspot.com/ |

## Antibodies

A monoclonal antibody against GluN1 (clone 54.1 MAB363) was obtained from Millipore (Germany), and its specificity was characterized previously (*Siegel et al., 1994*). An affinity-purified polyclonal rabbit anti-GluA1-4 (pan-AMPA), corresponding to aa 724–781 of rat, was used and characterized previously (*Nusser et al., 1998*).

## Immunohistochemistry for electron microscopy

Immunohistochemical reactions at the electron microscopic level were carried out using the post-embedding immunogold method as described earlier (*Lujan et al., 1996*). Briefly, animals (n = 3 rats) were anesthetized by intraperitoneal injection of ketamine-xylazine 1: 1 (0.1 mL/kg b.w.) and transcardially perfused with ice-cold fixative containing 4% paraformaldehyde, 0.1% glutaraldehyde, and 15% saturated picric acid solution in 0.1 M phosphate buffer (PB) for 15 min. Vibratome sections 500 μm thick were placed into 1 M sucrose solution in 0.1 M PB for 2 hr before they were slammed on a Leica EM CPC apparatus. Samples were dehydrated in methanol at −80°C and embedded by freeze-substitution (Leica EM AFS2) in Lowicryl HM 20 (Electron Microscopy Science, Hatfield, PA), followed by polymerization with UV light. Then, ultrathin 80-nm-thick sections from Lowicryl-embedded blocks of the hippocampus were picked up on coated nickel grids and incubated on drops of a blocking solution consisting of 2% human serum albumin in 0.05 M TBS and 0.03% Triton X-100. The grids were incubated with GluN1 or pan-AMPA antibodies (10 μg/mL in 0.05 M TBS and 0.03% Triton X-100 with 2% human serum albumin) at 28°C overnight. The grids were incubated on drops of goat anti-rabbit IgG conjugated to 10 nm colloidal gold particles (Nanoprobes Inc) in 2% human serum albumin and 0.5% polyethylene glycol in 0.05 M TBS and 0.03% Triton X-100. The grids were then washed in TBS and counterstained for electron microscopy with 1% aqueous uranyl acetate followed by Reynolds's lead citrate. Ultrastructural analyses were performed in a JEOL-1010 electron microscope.

## Hippocampal slice preparation

Animal handling followed an approved protocol by the Albert Einstein College of Medicine Institutional Animal Care and Use Committee in accordance with the National Institute of Health guidelines. Acute rat hippocampal slices (400 μm thick) were obtained from Sprague-Dawley rats, from postnatal day 17 (P17) to P28 of either sex. For procedures regarding transgenic mouse slice preparation, see below. The hippocampi were isolated and cut using a VT1200s microslicer (Leica Microsystems Co.) in a solution containing (in mM): 215 sucrose, 2.5 KCl, 26 NaHCO$_3$, 1.6 NaH$_2$PO$_4$, 1 CaCl$_2$, 4 MgCl$_2$, 4 MgSO$_4$, and 20 glucose. Acute slices were placed in a chamber containing a 1:1 mix of sucrose cutting solution and normal extracellular ACSF recording solution containing (in mM): 124 NaCl, 2.5 KCl, 26 NaHCO$_3$, 1 NaH$_2$PO$_4$, 2.5 CaCl$_2$, 1.3 MgSO$_4$, and 10 glucose incubated in a warm-water bath at 33–34°C. The chamber was brought to room temperature for at least 15 min post-sectioning, and the 1:1 sucrose-ACSF solution was replaced by ACSF. All solutions were equilibrated with 95% O$_2$ and 5% CO$_2$ (pH 7.4). Slices were allowed to recover for at least 45 min in the ACSF solution before recording. For physiological Mg$^{+2}$ and Ca$^{+2}$ experiments, ACSF solutions were adjusted to (in mM): 1.2 MgSO$_4$ and 1.2 CaCl$_2$, and temperature was maintained at 35 ± 0.1°C in the submersion-type recording chamber heated by a temperature controller (TC-344B Dual Automatic Temperature Controller, Warner Instruments).

## Electrophysiology

Electrophysiological recordings were performed at 26.0 ± 0.1°C (unless otherwise stated) in a submersion-type recording chamber perfused at 2 mL/min with normal ACSF supplemented with the GABA$_A$ receptor antagonist picrotoxin (100 μM) and the selective AMPA receptor (AMPAR) antagonist LY303070 at a low concentration (0.5 μM) to minimize CA3-CA3 recurrent activity, or at a high concentration (15 μM) to isolate KAR-EPSCs and KAR-EPSPs to assess monosynaptic mf transmission. Whole-cell recordings were made from CA3 pyramidal cells voltage-clamped at −70 mV using patch-type pipette electrodes (3–4 mΩ) containing (in mM): 131 cesium gluconate, 8 NaCl, 1 CaCl$_2$, 10 EGTA, 10 glucose, 10 HEPES, and 2 MK-801 pH 7.25 (280–285 mOsm) unless specified otherwise. KOH was used to adjust pH. Series resistance (8–15 MΩ) was monitored throughout all experiments with a −5 mV, 80 ms voltage step, and cells that exhibited a series resistance change (>20%) were excluded from analysis. A stimulating bipolar electrode (theta glass, Warner Instruments) was filled with ACSF and placed in *stratum lucidum* to selectively activate mfs using a DS2A Isolated Voltage Stimulator (Digitimer Ltd.) with a 100 μs pulse width duration. AMPAR-EPSCs were recorded for a baseline period of 2 min, and LFF was induced by stepping the stimulation frequency from 0.1 to 1 Hz for 2 min. Facilitation was measured by taking a ratio of the mean EPSC during the steady-state,

LFF period of activity and the 2-min baseline ($ESPC_{1Hz}/EPSC_{0.1Hz}$) before and after bath application of NMDAR antagonists.

To qualify for analysis, mf responses met three criteria: (1) The 20–80% rise time of the AMPAR-EPSC was less than 1 ms, (2) LFF was greater than 150%, (3) the AMPAR-EPSC displayed at least 70% sensitivity to the group 2/3 mGluR agonist, DCG-IV (1 µM). Isolated KAR-EPSCs were elicited by 2 pulses with a 5 ms inter-stimulus interval for LFF experiments. Baseline measurements were acquired at least 10 min after 'break-in' to achieve optimal intracellular blockade of postsynaptic NMDARs by MK-801 (2 mM) in the patch-pipette. To transect mf axons in acute slices, a 45° ophthalmic knife (Alcon Surgical) was used to make a diagonal cut across the hilus from the dorsal to ventral blades of the DG, and the subregion CA3b was targeted for patch-clamp recordings. For D-APV (2 mM) puff experiments, a puffer device (Toohey Company) was set to deliver two to three puffs of 100 ms duration at 3–4 psi during the 2 min of LFF activity. The puffer pipette was placed at least 200 µm away from the recording site, and both the puff pipette and hippocampal slice were positioned to follow the direction of the laminar perfusion flow in a low profile, submersion-type chamber (RC-26GLP, Warner Instruments). Burst-induced facilitation was elicited by 5 pulses at 25 Hz with a 0.03 Hz inter-trial interval for a baseline period of 10 min. Facilitation was measured by calculating the ratio of the mean KAR-EPSC peak of the fifth pulse to the first pulse (P5/P1) before and after bath application of MK-801 (50 µM). To study KAR induced action potentials, CA3 pyramidal cells were whole-cell patch-clamped with internal solution containing (in mM): 112 potassium gluconate, 17 KCl, 0.04 $CaCl_2$, 0.1 EGTA, 10 HEPES, 10 NaCl, 2 MgATP, 0.2 $Na_3$GTP, and 2 MK-801, pH 7.2 (280–285 mOsm). Current-clamped CA3 cells were held at −70 mV during burst stimulation of mfs (5 pulses at 25 Hz) to monitor evoked action potentials. Spike transfer was measured by quantifying mean number of spikes/burst for a 10 min period before and after bath application of MK-801 (50 µM). Robust sensitivity to the AMPAR/KAR selective antagonist NBQX (10 µM) confirmed KAR-EPSC responses. Similarly, CA3 pyramidal cells were kept in current-clamp mode for AMPAR-mediated action potential monitoring in the presence of LY303070 (0.5 µM) and picrotoxin (100 µM). AMPAR-mediated mf action potentials were confirmed by blockade of responses following application of DCG-IV (1 µM). Both hilar MCs and CA3 INs were visually patched-loaded with Alexa 594 (35 µM), and morphological identity was confirmed by two-photon laser microscopy at the end of experiments. MCs were voltage-clamped at −70 mV, and a bipolar electrode was placed in the DG to activate mf inputs. The data analysis and inclusion criteria used for mf experiments (described above) was also implemented for MC recordings. CA3 INs were voltage-clamped at −70 mV and burst stimulated, facilitation was assessed as previously mentioned. Both facilitating and depressing mf responses were included for analysis given the diversity of mf to CA3 IN transmission (*Toth et al., 2000*). Whole-cell voltage and current-clamp recordings were performed using an Axon MultiClamp 700B amplifier (Molecular Devices). Signals were filtered at 2 kHz and digitized at 5 kHz. Stimulation and acquisition were controlled with custom software (Igor Pro 7).

## Transgenic animals

*Grin1*-floxed littermate mice of either sex (P16-20) were injected with 1 µL of AAV5-CaMKII-eGFP, AAV5-CaMKII-CreGFP, AAV5-CaMKII-mCherry, or AAV5-CaMKII-mCherry-Cre viruses at a rate of 0.12 µL/min at coordinates (−1.9 mm A/P, 1.1 mm M/L, 2.4 mm D/V) targeting the DG using a stereotaxic apparatus (Kopf Instruments). Two weeks post-surgery, mice were sacrificed for electrophysiology or $Ca^{2+}$ imaging experiments. Mice were transcardially perfused with 20 mL of cold NMDG solution containing (in mM): 93 NMDG, 2.5 KCl, 1.25 $NaH_2PO_4$, 30 $NaHCO_3$, 20 HEPES, 25 glucose, 5 sodium ascorbate, 2 Thiourea, 3 sodium pyruvate, 10 $MgCl_2$, 0.5 $CaCl_2$, brought to pH 7.35 with HCl. The hippocampi were isolated and cut using a VT1200s microslicer in cold NMDG solution. Acute mouse slices were placed in an incubation chamber containing normal ACSF solution that was kept in a warm-water bath at 33–34°C. All solutions were equilibrated with 95% $O_2$ and 5% $CO_2$ (pH 7.4). Post-sectioning, slices recovered at room temperature for at least 45 min prior to experiments. For NMDAR/AMPAR ratios, GCs were patch-clamped with the cesium internal solution previously mentioned containing either Alexa 594 (35 µM) for GFP+ cells (laser tuned to 830 nm/910 nm, respectively) or Alexa 488 (35 µM) for mCherry+ cells (laser tuned to 910 nm/780 nm, respectively). AMPAR-EPSCs were recorded at −65 mV in the presence of picrotoxin (100 µM) by placing a bipolar electrode near the medial perforant path and delivering a 100 µs pulse width duration using an Iso-flex stimulating unit. AMPAR-EPSCs were acquired for at least 5 min followed by bath application of

NBQX (10 µM) to isolate NMDAR-EPSCs. GCs were brought to +40 mV to alleviate magnesium block and record optimal NMDAR-EPSCs. NMDAR/AMPAR ratios were measured by taking the mean NMDAR-EPSC/AMPAR-EPSC for a 5 min period of each component. Only acute mouse slices with optimal GFP and mCherry reporter fluorescence (i.e. robust expression, ≥75% of DG fluorescence) were used for electrophysiology, and $Ca^{2+}$ and BDNF imaging experiments. *Grin1*-floxed animals (The Jackson Laboratory) were kindly provided by Dr. Michael Higley (Yale University).

## Optogenetics

*Grin1* floxed and control mice of either sexes (P17–P20) were injected with a 1:2 mix of AAV5-CaM-KII-CreGFP/AAV-DJ-FLEX-ChIEF-tdTomato viruses targeting the DG, using the same coordinates described above. At least 4 weeks post-surgery, acute hippocampal slices were prepared as previously described, and slices showing optimal GFP and tdTomato expression were used for electrophysiology experiments. Mf optical burst stimulation was elicited by using a Coherent 473 nm laser (4–8 mW) delivering 5 pulses at 25 Hz with a 1–2 ms pulse width duration. Facilitation was measured by taking a ratio of the mean AMPAR-EPSC peak of the fifth pulse to the first pulse (P5/P1) in control and *Grin1*-cKO animals.

## Two-photon calcium imaging and MNI-glutamate uncaging

mCherry$^+$ GCs were patch-loaded with an internal solution containing in (mM): 130 KMeSO$_4$, 10 HEPES, 4 MgCl$_2$, 4 Na$_2$ATP, 0.4 NaGTP, 10 sodium phosphocreatine, 0.035 Alexa 594 (red morphological dye), and 0.2 Fluo-5F (green calcium indicator), 280–285 mOsm. KOH was used to adjust pH. GCs near the hilar border were avoided and GCs that exhibited adult-born GC electrophysiological properties were excluded from analysis. GCs were kept in voltage clamp configuration at −50 mV for at least 1 hr to allow the diffusion of dyes to mf boutons. Recordings were obtained in ACSF solution containing (in mM): 124 NaCl, 2.5 KCl, 26 NaHCO$_3$, 1 NaH$_2$PO$_4$, 4 CaCl$_2$, 0 MgSO$_4$, 10 glucose, 0.01 NBQX, 0.1 picrotoxin, and 0.01 D-serine. Using an Ultima 2P laser scanning microscope (Bruker Corp) equipped with an Insight Deep See laser (Spectra Physics) tuned to 830 nm, the 'red' photomultiplier tube (PMT) was turned on and with minimal pockel power the red signal was used to identify the mf axon. With 512 × 512 pixel resolution, mf axons were followed for at least 200 µm from the DG toward CA3, until bouton structures were morphologically identified and measured (>3 µm in diameter). GCs were switched to current-clamp mode held at −70 mV and 1 ms current injections were used to elicit a burst of 5 action potentials at 25 Hz. Using line scan analysis software (PrairieView 5.4, Bruker Corp.), a line was drawn across the diameter of the bouton at a magnification of at least 16×. The 'green' PMT channel was turned on, and 1000 line scans were acquired in a 2 s period. Action potential induction was delayed for 400 ms to collect a baseline fluorescence time period. Calcium transients (CaTs) were acquired with a 1 min inter-trial-interval and analyzed using the ∆G/R calculation: $(G − G_0)/R$. CaTs from control animals were compared to *Grin1*-cKO by taking the mean peak ∆G/R value for a 30 ms period of the fifth action potential. In similar fashion, CaT signals in acute rat hippocampal slices were acquired and tested for sensitivity to D-APV (100 µM) while adjusting ACSF MgSO$_4$ concentration to 1.3 mM and CaCl$_2$ to 2.5 mM in the absence of NBQX.

For glutamate uncaging experiments, GCs that were mCherry$^+$ were patch-loaded using the internal solution previously described, and a small volume (12 mL) of recirculated ACSF solution containing (in mM): 124 NaCl, 2.5 KCl, 26 NaHCO$_3$, 1 NaH$_2$PO$_4$, 4 CaCl$_2$, 0 MgSO$_4$, 10 glucose, 2.5 MNI-glutamate, 0.01 NBQX, 0.1 picrotoxin, and 0.01 D-serine. A MaiTai HP laser (Spectra Physics) was tuned to 720 nm to optimally uncage glutamate and elicit CaTs in GC dendritic spines. Following the measurement of CaTs in GC spines, mf boutons were identified and to mimic bursting activity, five uncaging pulses (1 ms duration) were delivered at 25 Hz. The acquired CaTs in spines and boutons were analyzed using the ∆G/R calculation in control and *Grin1*-cKO animals. In a subset of control boutons, D-APV (100 µM) was applied to detect CaT sensitivity to NMDAR antagonism.

## Two-photon BDNF-phluorin imaging

*Grin1* floxed and control mice of both sexes (P16-20) were injected with a 1:2 mix of AAV5-CaMKII-mCherryCre/AAV-DJ-DIO-BDNF-phluorin viruses targeting the DG using the same coordinates as above. At least 4 weeks post-surgery, acute hippocampal slices were prepared as previously described, and slices showing optimal GFP and mCherry expression were taken for imaging sessions.

For stimulation, a monopolar micropipette electrode was placed in the *stratum lucidum* at least 250 µm away from the imaging site. The Insight Deep See laser (Spectra Physics) was tuned to 880 nm, and the imaging site was selected by the appearance of fibers and bouton structures in the *stratum lucidum*. Using 512 × 512 pixel resolution identified boutons measuring at least 3 µm in diameter were selected as a region of interest (ROI) magnified to 4–6×, and a baseline acquisition of 100 consecutive images at 1 Hz using T-series software (PrairieView 5.4, Bruker Corp.) was acquired (*Park et al., 2014*). Following the baseline acquisition, a repetitive stimulation consisting of 125 pulses at 25 Hz was delivered 2×, triggering an acquisition of 200 consecutive images at 1 Hz. The fluorescence intensity of the bouton ROI was measured using ImageJ software to calculate ΔF/F of the BDNF-pHluorin signal. To verify reactivity of the ROI, an isosmotic solution of $NH_4Cl$ (50 mM) was added at the end of the imaging session as previously reported (*Park et al., 2014*). The same experimental and analysis procedure was implemented to measure BDNF release triggered by mf burst stimulation consisting of 5 pulses at 100 Hz, 50×, every 0.5 s.

## Viruses

AAV5-CaMKII-eGFP and AAV5-CaMKII-CreGFP viruses were acquired from UPenn Vector Core. AAV5-CaMKII-mCherry and AAV5-CaMKII-mCherry-Cre were obtained from UNC Chapel Hill Vector Core. The AAV-DJ-FLEX-ChIEF-tdTomato and AAV-DJ-DIO-BDNF-phluorin viruses were custom ordered and obtained from UNC Chapel Hill Vector Core. The DNA of the ChIEF virus was a generous gift from Dr. Pascal Kaeser (Harvard University), and the DNA of the BDNF-pHluorin was kindly provided by Dr. Hyungju Park (Korea Brain Research Institute).

## Chemicals and drugs

Picrotoxin and all chemicals used to prepare cutting, recording, and internal solutions were acquired from MilliporeSigma. All NMDAR antagonists (D-APV, MK-801, R-CPP), NMDAR agonist (D-serine), the group 2/3 mGluR agonist (DCG-IV), and MNI-glutamate for uncaging experiments were purchased from Tocris Bioscience. D-APV was also acquired from the NIMH Chemical Synthesis Drug Program. NBQX was purchased from Cayman Chemical Company. The noncompetitive AMPAR selective antagonist LY303070 was custom ordered from ABX Chemical Company. Alexa 594 morphological dye, Alexa 488, and the $Ca^{2+}$ indicator Fluo-5F (Invitrogen) were purchased from Thermo-Fisher Scientific.

## Statistical analysis and data acquisition

All data points from experiments were tested for normality using a Shapiro–Wilk test (p-value < 5% for a normal distribution). Statistical significance was determined if p-value < 0.05. Experiments with a normal distribution and an N > 7 cells were tested for statistical significance with a paired Student's t-test. Experiments with N < 7 cells or skewed distributions were tested for statistical significance using a paired Wilcoxon signed-rank sum test. For experiments comparing control and *Grin1*-cKO animals, statistical significance was determined using unpaired t-test and Mann–Whitney test (U < 0.05). All statistical tests were performed using Origin Pro 9 (Origin Lab). Experimenters were blind to the identity of the virus injected in transgenic *Grin1* floxed mice during the acquisition of data in CA3 electrophysiology and two-photon imaging. However, data analysis could not be performed blind in those experiments in which NMDAR/AMPAR ratios in GCs were examined in order to assess the efficiency of the cKO.

## Acknowledgements

We thank all the Castillo lab members for invaluable discussions. We also thank Dr. Hyungju Park for his generous gift of the BDNF-phluorin DNA construct, Dr. Michael Higley for sharing *Grin1* floxed mice, and Dr. Pascal Kaeser for his generous gift of the Cre-dependent ChIEF DNA construct. Funding sources: This work supported by the NIH (F31-MH109267 to PJL; R01 MH116673, R01MH125772, and R01 NS 113600 to PEC) and by the Spanish Ministerio de Economia y Competitividad (RTI2018-095812-B-I00) and Junta de Comunidades de Castillo-La Mancha (SBPLY/17/180501/000229) to RL.

## Additional information

### Funding

| Funder | Grant reference number | Author |
|---|---|---|
| National Institutes of Health | R01 MH116673 | Pablo E Castillo |
| National Institutes of Health | R01 MH125772 | Pablo E Castillo |
| National Institutes of Health | R01 NS113600 | Pablo E Castillo |
| National Institutes of Health | F31 MH 109267 | Pablo J Lituma |

The funders had no role in study design, data collection and interpretation, or the decision to submit the work for publication.

### Author contributions

Pablo J Lituma, Conceptualization, Data curation, Formal analysis, Funding acquisition, Investigation, Methodology, Writing - original draft, Writing - review and editing; Hyung-Bae Kwon, Conceptualization, Investigation, Methodology, Writing - review and editing; Karina Alviña, Validation, Methodology, Writing - review and editing; Rafael Luján, Data curation, Formal analysis, Investigation, Methodology, Writing - review and editing; Pablo E Castillo, Conceptualization, Resources, Supervision, Funding acquisition, Methodology, Writing - original draft, Project administration, Writing - review and editing

### Author ORCIDs

Pablo J Lituma (iD) http://orcid.org/0000-0001-8442-3622
Pablo E Castillo (iD) https://orcid.org/0000-0002-9834-1801

### Ethics

Animal experimentation: Animal handling followed a protocol approved by the Albert Einstein College of Medicine Institutional Animal Care and Use Committee (IACUC protocols 00001043, 00001047 and 00001053) in accordance with National Institute of Health guidelines.

### Decision letter and Author response

Decision letter https://doi.org/10.7554/eLife.66612.sa1
Author response https://doi.org/10.7554/eLife.66612.sa2

## Additional files

### Supplementary files

- Source data 1. Source datasets for all figures.

- Transparent reporting form

### Data availability

All data generated or analyzed during this study are included in the manuscript and supporting files.

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
