## [Decision Letter]

**Acceptance summary:**

This paper demonstrates functional presynaptic NMDA receptors at mossy fiber terminals in the hippocampus. Postsynaptic NMDA receptors are critically involved in learning and memory as coincidence detectors in Hebbian plasticity. Some studies, however, have reported that NMDA receptors may function in more unconventional manners. This paper provides strong evidence for presynaptic NMDA receptors at a specific subset of hippocampal mossy-fibre boutons. Electron microscopy, electrophysiology, optogenetics, calcium imaging, and genetic manipulation yield compelling evidence that supports the main conclusions.

**Decision letter after peer review:**

Thank you for submitting your article "Presynaptic NMDA receptors facilitate short-term plasticity and BDNF release at hippocampal mossy fiber synapses" for consideration by *eLife*. Your article has been reviewed by 3 peer reviewers, one of whom is a member of our Board of Reviewing Editors, and the evaluation has been overseen by Gary Westbrook as the Senior Editor. The following individuals involved in review of your submission have agreed to reveal their identity: Per Jesper Sjöström (Reviewer #2); Kenneth A Pelkey (Reviewer #3).

Essential revisions:

1) Experimental:

Re-evaluation of the preNMDAR blockade effect on frequency facilitation under more physiological [Ca^2+^]o (1.2 mM) and temperature is essential given the precedents for unique modes of mossy fiber release with unique Ca^2+^ source dependencies at variable [Ca^2+^]o levels and temperatures. Since change in cleft glutamate concentration could impact how and when presynaptic receptors are activated, this new data would be important to establish the physiological relevance of the authors' findings.

While re-evaluation of the calcium imaging experiments in physiological temperature and divalent concentration is not required. Please, provide a thorough and careful discussion on the limitation of the experimental conditions used in this study.

2) Explanation:

The authors convincingly demonstrate the involvement of preNMDARs in both LFF and burst facilitation at mossy fiber synapses. While the proposed mechanism for preNMDAR activation during burst facilitation is fairly straightforward, it is less clear how the requirements for ionotropic NMDAR activation are met during low-frequency 1 Hz stimulation. Please, comment on how the glutamate from a presynaptic spike can activate preNMDARs at 1 Hz when the depolarization from that spike is gone. Although this could work at some higher frequency, when the subsequent spikes in a burst provide the necessary depolarization, it is not clear how this would work at 1 Hz.

The mechanism of relief for preNMDAR voltage dependent block needs to be thoroughly discussed (but not necessarily experimentally solved). MFB recordings reveal APs with sub ms half durations even following use dependent spike broadening, this duration makes it difficult to expect that the presynaptic spikes themselves can support depolarization of sufficient duration to relieve the block. These MFB spikes do exhibit ADPs that could sum but published traces (at 5Hz MFB APs) do not support significant summated depolarization of the ADPs within the terminal (Geiger and Jonas, Neuron 2000).

3) Clarification:

In the calcium imaging experiments signal-to-noise ratio appears to be poor on Figure 5, 6, S5 and S6, where responses are typically well below 5%. This is echoed by the fuzzy Fluo5-F example images, making conclusions drawn from the data not particularly strong. In some cases, perhaps the wrong images were shown? Maybe images did not render correctly in the PDF I look at? Please clarify.

*Reviewer #1 (Recommendations for the authors):*

1. The immunocytochemistry images are blurry; the synaptic vesicles are not clearly visible in the presynaptic terminal. It would be great to provide better quality illustrations for these experiments.

2. Optogenetic experiments shown in Figure 4A-C demonstrating a role for preNMDAR in short-term facilitation: The authors demonstrate that Grin1-cKO decreases P5/P1. The narrative suggests that this change should be attributed to a decrease in P5. However, in the example shown, P5 appears similar in control and in Grin1-cKO, while P1 appears to be increased in Grin1-cKO. Are there changes in basal release in Grin1-cKO animals?

3. From the images presented in Figure 5B, it is hard to evaluate where the boutons are recorded from.

4. For both uncaging and Ca^2+^ imaging experiments, data recorded in control mice is compared to data recorded in Grin1-cKO animals. Pharmacological blockade of NMDARs in the same boutons would provide more insight on the relative contribution of preNMDAR to presynaptic Ca^2+^ transients evoked by somatic APs or glutamate uncaging pulses.

5. Is there a special relationship between NMDAR and BDNF release? Or is it just that Grin1-cKO boutons experience a lower total Ca^2+^ influx during the MF stimulation paradigm?

*Reviewer #2 (Recommendations for the authors):*

This manuscript is succinct and well-written, making it a pleasure to read. A caveat of the study is that the imaging experiments presented appear to have very low signal:noise, preventing convincing conclusions to be drawn. In addition, while the BDNF finding is potentially important and supports the presence of preNMDARs, it seems to be largely disconnected from the rest of the story. Finally, it is not clear how preNMDAR autoreceptors can signal ionotropically at 1 Hz. These issues are elaborated in the points below. Nonetheless, these issues do not affect the overall conclusions of the paper, which is well-grounded with good experimental design and execution. We believe this paper should be highly suitable for publication in *eLife* after these points have been addressed.

1. BDNF: The finding that preNMDARs contribute to BDNF release is very intriguing. However, it seems to be just loosely linked to the rest of the story. Could this be tied in better somehow? In Figure 7, the authors elicit BDNF release through a repeated "burst" stimulation of 125 pulses at 25 Hz. I think the use of the word "burst" for this kind of sustained stimulation is misleading, especially in comparison with previous figures where burst stimulation consisted of 5 pulses. I also wonder why the authors used this form of stimulation, as opposed other stimulation protocols like TBS, which is both effective at eliciting BDNF release (Balkowiec and Katz, 2002) and more closely mimics GCs' sparse, bursting activity in vivo (Pernia-Andrade and Jonas, 2014). In Figure 7, if Grin1-cKO reduces BDNF release physiologically, one would expect the baseline BDNF-pHluorin signal to be significantly higher in the cKO compared to the control. Has this been compared?

2. Statistics and Controls: In Figure 8, unlike in previous figures, it is not shown whether controls were done to check for stability of responses over time, either in interneurons or hilar mossy cells. This is particularly missed in 8B, as s. lucidum interneurons can show synapse-type specific long-term plasticity that affects burst facilitation (Toth et al., 2000). The mixed responses shown in 8C may reflect the synaspe dichotomy shown by Toth et al., and it could be difficult to conclude about the role of preNMDARs at interneuron synapses without further exploration of these differences. The paired t-tests used throughout the paper provide a powerful internal comparison (Figure 1, S2, 3, 4, 8). However, as these experiments involves two rounds of LFF induction over time, drug treatment is not the only variable. Dialysis of cells after gaining whole-cell access, potential changes in efficacy of consecutive LFF induction and cell death after axotomy (Figure 3, S4), for example, can also have large influences on the results. Therefore, naive/solvent controls (like Figure S2, 3B, 3D, 4E, 4G) should have been done for each set of experiments and compared statistically with the drug treatment groups (i.e. After/before of control vs. after/before of drug treatment groups with one-way ANOVA or equivalent tests). N numbers were given in boutons/spines. It was unclear how many cells/slices/biological repeats were performed. The n=6-10 spines/boutons seem rather small. Please clarify.

3. Lines 265-266, this seems like an erroneous conclusion to me: "Thus, preNMDARs contribute significantly to presynaptic Ca^2+^ rise in mossy fiber boutons, and by this means facilitate synaptic transmission." Indirect action of preNMDARs on transmission is still a possibility, even if presynaptic calcium increases when preNMDARs are activated, no? That calcium goes up does not mean that this is how the preNMDARs act, it just means it is a possible route of action. Please clarify.

*Reviewer #3 (Recommendations for the authors):*

In this manuscript Lituma and colleagues describe a role for presynaptic NMDARs at hippocampal mossy fiber (MF) synapses in activity dependent short-term plasticity of release onto CA3 pyramid and mossy cell postsynaptic targets but not at MF-interneuron synapses. The combined use of electron microscopy, electrophysiological, optogenetic, calcium imaging, and genetic manipulation approaches expertly employed by the authors yields high quality compelling evidence in full support of the study's main conclusions. Overall, the investigation is well designed with a clear hypothesis, appropriate methodological considerations, and logical flow resulting in a well written manuscript that is sure to be of broad scientific interest. However, I do have three major points for consideration to improve the manuscript and further ensure the physiological relevance of the findings.

1) The methods state that all electrophysiological assays were performed at 26 degrees

Celsius. Hypothermic conditions can suppress transmitter uptake and promote glutamate pooling/spillover for activation of presynaptic receptors capable of modulating release that is not readily apparent at physiological temperatures (Min et al., 1998). It seems important therefore that the authors confirm the ability of presynaptic NMDARs to contribute to short term facilitation of MF-CA3 pyramid transmission at physiological temperatures.

2) The data fully support that presynaptic NMDARs have the capacity to contribute to presynaptic calcium transients (CaTs) and enhanced transmitter release. However, left undetermined is whether presynaptic NMDAR-mediated calcium events alone can promote vesicle fusion and release or if they can only enhance release over and above that initially triggered by CaTs from activation of voltage gated calcium channels (VGCCs). A potential role for presynaptic NMDARs in driving spontaneous action potential independent release at MF synapses is alluded to in the discussion. In recordings with intracellular MK-801 (with or without extracellular TTX) does subsequent NMDAR blockade alter spontaneous event frequency or is spontaneous frequency measurably reduced following loss of GRIN1 in granule cells? Of note on this subject combined blockade of P/Q- and N-type VGCCs appears to entirely eliminate MF-CA3 transmission probed with short train stimulation at comparable frequencies to the current study (Chamberland et al., 2020).

3) The presynaptic calcium imaging experiments provide convincing evidence for CaTs mediated by presynaptic NMDARs. However, the physiologically relevant capacity for similar NMDAR-mediated CaTs is hard to estimate as the imaging experiments were performed in the absence of magnesium. It would of interest to know if presynaptic NMDARs have unique magnesium sensitivity or if voltage-dependent block can be overcome during brief train stimulation.

---

## [Author Response]

Essential revisions:1) Experimental:Re-evaluation of the preNMDAR blockade effect on frequency facilitation under more physiological [Ca^2+^]o (1.2 mM) and temperature is essential given the precedents for unique modes of mossy fiber release with unique Ca^2+^ source dependencies at variable [Ca^2+^]o levels and temperatures. Since change in cleft glutamate concentration could impact how and when presynaptic receptors are activated, this new data would be important to establish the physiological relevance of the authors' findings.While re-evaluation of the calcium imaging experiments in physiological temperature and divalent concentration is not required. Please, provide a thorough and careful discussion on the limitation of the experimental conditions used in this study.

We thank the reviewers for raising this important point regarding physiological temperature and divalent concentration. We have performed new experiments at 35ºC and 1.2 mM Ca^+2^ and 1.2 mM Mg^2+^ extracellular concentration and present our findings in Figure 4—figure supplement 1. Under these more physiological experimental conditions, we show that preNMDARs contribute to burst-induced facilitation.

2) Explanation:The authors convincingly demonstrate the involvement of preNMDARs in both LFF and burst facilitation at mossy fiber synapses. While the proposed mechanism for preNMDAR activation during burst facilitation is fairly straightforward, it is less clear how the requirements for ionotropic NMDAR activation are met during low-frequency 1 Hz stimulation. Please, comment on how the glutamate from a presynaptic spike can activate preNMDARs at 1 Hz when the depolarization from that spike is gone. Although this could work at some higher frequency, when the subsequent spikes in a burst provide the necessary depolarization, it is not clear how this would work at 1 Hz.The mechanism of relief for preNMDAR voltage dependent block needs to be thoroughly discussed (but not necessarily experimentally solved). MFB recordings reveal APs with sub ms half durations even following use dependent spike broadening, this duration makes it difficult to expect that the presynaptic spikes themselves can support depolarization of sufficient duration to relieve the block. These MFB spikes do exhibit ADPs that could sum but published traces (at 5Hz MFB APs) do not support significant summated depolarization of the ADPs within the terminal (Geiger and Jonas, Neuron 2000).

We are pleased the reviewers note we convincingly demonstrate the involvement of preNMDARs in both LFF and burst facilitation in mossy fiber synapses. We also wondered about the mechanism underlying preNMDAR activation at 1 Hz. In our revised manuscript (Lines 367-378), we attempted an explanation as follows: “There is evidence that preNMDARs can operate as coincidence detectors at some synapses (Duguid, 2013; Wong et al., 2020). […] By alleviating the magnesium blockade, these potentials might reduce the need for coincidence detection and transiently boost the functional impact of mf preNMDARs.”

3) Clarification:In the calcium imaging experiments signal-to-noise ratio appears to be poor on Figure 5, 6, S5 and S6, where responses are typically well below 5%. This is echoed by the fuzzy Fluo5-F example images, making conclusions drawn from the data not particularly strong. In some cases, perhaps the wrong images were shown? Maybe images did not render correctly in the PDF I look at? Please clarify.

We thank the reviewer for this observation. In response, we have replaced the images and ensured they render correctly in PDF format.

Reviewer #1 (Recommendations for the authors):1. The immunocytochemistry images are blurry; the synaptic vesicles are not clearly visible in the presynaptic terminal. It would be great to provide better quality illustrations for these experiments.

Agreed. We have replaced the images with improved quality in Figure 1A-C.

2. Optogenetic experiments shown in Figure 4A-C demonstrating a role for preNMDAR in short-term facilitation: The authors demonstrate that Grin1-cKO decreases P5/P1. The narrative suggests that this change should be attributed to a decrease in P5. However, in the example shown, P5 appears similar in control and in Grin1-cKO, while P1 appears to be increased in Grin1-cKO. Are there changes in basal release in Grin1-cKO animals?

We provide more representative traces in Figure 4B. We found no significant differences in basal transmitter release, as indicated by a comparable paired-pulse ratio as stated in the manuscript (Line 201).

3. From the images presented in Figure 5B, it is hard to evaluate where the boutons are recorded from.

We thank the reviewer for this observation. We have replaced the images and provided clearer examples of boutons in Figure 5B.

4. For both uncaging and Ca^2+^ imaging experiments, data recorded in control mice is compared to data recorded in Grin1-cKO animals. Pharmacological blockade of NMDARs in the same boutons would provide more insight on the relative contribution of preNMDAR to presynaptic Ca^2+^ transients evoked by somatic APs or glutamate uncaging pulses.

We have performed the requested experiments and assessed CaTs evoked by somatic APs (Figure 5—figure supplement 1) and glutamate uncaging pulses (Figure 6—figure supplement 3) before and after NMDAR antagonism with D-APV.

5. Is there a special relationship between NMDAR and BDNF release? Or is it just that Grin1-cKO boutons experience a lower total Ca^2+^ influx during the MF stimulation paradigm?

The precise relationship between NMDAR and BDNF release remains poorly understood. A previous study suggested presynaptic Ca^+2^ influx via preNMDARs during repetitive stimulation, together with calcium released from internal stores, contributes to BDNF release at corticostriatal synapses (Park et al., Neuron 2014). While we have not measured Ca^+2^ influx during our MF stimulation paradigm, our observations are consistent with reductions in presynaptic Ca^+2^ influx underlying diminished BDNF release in *Grin1*-cKO boutons, and we do not discard the potential contribution of internal calcium stores.

Reviewer #2 (Recommendations for the authors):This manuscript is succinct and well-written, making it a pleasure to read. A caveat of the study is that the imaging experiments presented appear to have very low signal:noise, preventing convincing conclusions to be drawn. In addition, while the BDNF finding is potentially important and supports the presence of preNMDARs, it seems to be largely disconnected from the rest of the story. Finally, it is not clear how preNMDAR autoreceptors can signal ionotropically at 1 Hz. These issues are elaborated in the points below. Nonetheless, these issues do not affect the overall conclusions of the paper, which is well-grounded with good experimental design and execution. We believe this paper should be highly suitable for publication in eLife after these points have been addressed.1. BDNF: The finding that preNMDARs contribute to BDNF release is very intriguing. However, it seems to be just loosely linked to the rest of the story. Could this be tied in better somehow? In Figure 7, the authors elicit BDNF release through a repeated "burst" stimulation of 125 pulses at 25 Hz. I think the use of the word "burst" for this kind of sustained stimulation is misleading, especially in comparison with previous figures where burst stimulation consisted of 5 pulses. I also wonder why the authors used this form of stimulation, as opposed other stimulation protocols like TBS, which is both effective at eliciting BDNF release (Balkowiec and Katz, 2002) and more closely mimics GCs' sparse, bursting activity in vivo (Pernia-Andrade and Jonas, 2014).

The 125-pulse, 25 Hz stimulation protocol is commonly used to induce LTP at the mossy fiber to CA3 pyramidal cell synapse. Given that LTP at this synapse requires BDNF release, we decided to use this protocol first. We agree with the reviewer that TBS patterns of activity more closely mimic GC bursting activity in vivo. New experiments, now included in Figure 7—figure supplement 1, showed that BDNF release by more physiological burst stimulation is also reduced in the absence of preNMDARs.

In Figure 7, if Grin1-cKO reduces BDNF release physiologically, one would expect the baseline BDNF-pHluorin signal to be significantly higher in the cKO compared to the control. Has this been compared?

We have compared the baseline BDNF-pHluorin raw signals in Control and *Grin1*-cKO and found no significant difference (Control: 353.6 ± 72, n = 12 slices; *Grin1*-cKO: 286.6 ± 29, n = 10 slices; p = 0.435, unpaired *t*-test). Our findings suggest that preNMDARs facilitate BDNF release in an activity-dependent manner.

2. Statistics and Controls: In Figure 8, unlike in previous figures, it is not shown whether controls were done to check for stability of responses over time, either in interneurons or hilar mossy cells. This is particularly missed in 8B, as s. lucidum interneurons can show synapse-type specific long-term plasticity that affects burst facilitation (Toth et al., 2000). The mixed responses shown in 8C may reflect the synaspe dichotomy shown by Toth et al., and it could be difficult to conclude about the role of preNMDARs at interneuron synapses without further exploration of these differences.

We have added the stability experiments for CA3 interneurons and hilar mossy cells that we did not include in the original submission (see Figure 8—figure supplement 1). In response to the reviewer’s comment regarding facilitating and depressing CA3 inhibitory neurons, our data is now split into two groups i.e. facilitating and depressing synaptic inputs. NMDAR antagonism still had no effect on either population (Figure 8B).

The paired t-tests used throughout the paper provide a powerful internal comparison (Figure 1, S2, 3, 4, 8). However, as these experiments involves two rounds of LFF induction over time, drug treatment is not the only variable. Dialysis of cells after gaining whole-cell access, potential changes in efficacy of consecutive LFF induction and cell death after axotomy (Figure 3, S4), for example, can also have large influences on the results. Therefore, naive/solvent controls (like Figure S2, 3B, 3D, 4E, 4G) should have been done for each set of experiments and compared statistically with the drug treatment groups (i.e. After/before of control vs. after/before of drug treatment groups with one-way ANOVA or equivalent tests). N numbers were given in boutons/spines. It was unclear how many cells/slices/biological repeats were performed. The n=6-10 spines/boutons seem rather small. Please clarify.

The design of most of our experiments included internal controls. We understand this approach is one of the best ways to deal with variability across experiments. While running two consecutive rounds of LFF (with or without axotomy), could affect the magnitude of facilitation, we did not observe any significant change in naïve conditions.

We have revised the Figure Legends to clarify the number of animals, slices, cells, spines, or boutons.

Maintaining GCs patch-loaded for >1 hr while recirculating uncaging solutions were low yield experiments; 6 spines or 10 boutons were the highest numbers of experiments we could achieve to perform acceptable statistical analysis.

3. Lines 265-266, this seems like an erroneous conclusion to me: "Thus, preNMDARs contribute significantly to presynaptic Ca^2+^ rise in mossy fiber boutons, and by this means facilitate synaptic transmission." Indirect action of preNMDARs on transmission is still a possibility, even if presynaptic calcium increases when preNMDARs are activated, no? That calcium goes up does not mean that this is how the preNMDARs act, it just means it is a possible route of action. Please clarify.

We have no evidence for a preNMDAR-mediated, Ca^2+^ rise-independent effect on synaptic transmission. In any case, in response to the reviewer’s suggestion, we have modified the sentence as follows: “Thus, preNMDARs contribute significantly to presynaptic Ca^2+^ rise in mossy fiber boutons, and by this means likely facilitates synaptic transmission, although a potential contribution of Ca^2+^ rise-independent effects cannot be discarded.” (Lines 273-275).

Reviewer #3 (Recommendations for the authors):In this manuscript Lituma and colleagues describe a role for presynaptic NMDARs at hippocampal mossy fiber (MF) synapses in activity dependent short-term plasticity of release onto CA3 pyramid and mossy cell postsynaptic targets but not at MF-interneuron synapses. The combined use of electron microscopy, electrophysiological, optogenetic, calcium imaging, and genetic manipulation approaches expertly employed by the authors yields high quality compelling evidence in full support of the study's main conclusions. Overall, the investigation is well designed with a clear hypothesis, appropriate methodological considerations, and logical flow resulting in a well written manuscript that is sure to be of broad scientific interest. However, I do have three major points for consideration to improve the manuscript and further ensure the physiological relevance of the findings.1) The methods state that all electrophysiological assays were performed at 26 degreesCelsius. Hypothermic conditions can suppress transmitter uptake and promote glutamate pooling/spillover for activation of presynaptic receptors capable of modulating release that is not readily apparent at physiological temperatures (Min et al., 1998). It seems important therefore that the authors confirm the ability of presynaptic NMDARs to contribute to short term facilitation of MF-CA3 pyramid transmission at physiological temperatures.

The reviewer raises an important point regarding physiological temperature and glutamate uptake. In response, we have performed new experiments at more physiological recording conditions: 35 ºC, and 1.2 mM Ca^+2^ and 1.2 mM Mg^2+^ extracellular concentrations. Our new results presented in Figure 4—figure supplement 1 confirm that preNMDARs contribute to short-term plasticity of mf to CA3 pyramidal cell synaptic transmission at a physiological temperature, and Ca^2+^ and Mg^2+^ extracellular concentrations.

2) The data fully support that presynaptic NMDARs have the capacity to contribute to presynaptic calcium transients (CaTs) and enhanced transmitter release. However, left undetermined is whether presynaptic NMDAR-mediated calcium events alone can promote vesicle fusion and release or if they can only enhance release over and above that initially triggered by CaTs from activation of voltage gated calcium channels (VGCCs). A potential role for presynaptic NMDARs in driving spontaneous action potential independent release at MF synapses is alluded to in the discussion. In recordings with intracellular MK-801 (with or without extracellular TTX) does subsequent NMDAR blockade alter spontaneous event frequency or is spontaneous frequency measurably reduced following loss of GRIN1 in granule cells? Of note on this subject combined blockade of P/Q- and N-type VGCCs appears to entirely eliminate MF-CA3 transmission probed with short train stimulation at comparable frequencies to the current study (Chamberland et al., 2020).

The reviewer raises another important question, namely, whether preNMDAR-mediated Ca^+2^ is sufficient to promote transmitter release. While we have no evidence for or against this possibility, direct demonstration likely requires uncaging NMDA onto identified presynaptic boutons in the presence of a cocktail of VGCC blockers. We did not pursue this avenue given the high cost and low benefit ratio of these experiments. As denoted by the reviewer, Chamberland et al., 2020 demonstrated P/Q and N-type VGCC blockade entirely eliminates MF-CA3 transmission –also reported in the Castillo et al., 1994 study. These studies strongly suggest that the bulk of presynaptic Ca^2+^ rise that triggers neurotransmitter release is mediated by VGCCs, suggesting that preNMDARs mainly play a regulatory role by boosting release.

As for a potential role of preNMDARs in facilitating spontaneous AP-independent release, elucidating such role is not straightforward given that mossy fiber inputs comprise a small fraction of the excitatory synapses impinging on a CA3 pyramidal neuron. As a result, a potential reduction in mEPSC activity by NMDAR antagonism (or genetic *Grin1* removal from GCs) is likely to be lost in the background activity. In this context, it is worth noting that consistent with previous reports (Kamiya and Ozawa, J Physiol 1999; Kamiya et al., J Physiol 1996), we have evidence that the mGluR2/3 agonist DCG-IV (1-2 µM), which virtually abolishes evoked mossy fiber transmission, has a minimal effect on mEPSC activity in CA3 pyramidal cells. Thus, there are little reasons to believe that a significant reduction in mEPSC activity could be detected in CA3 pyramidal neurons following NMDAR antagonism.

3) The presynaptic calcium imaging experiments provide convincing evidence for CaTs mediated by presynaptic NMDARs. However, the physiologically relevant capacity for similar NMDAR-mediated CaTs is hard to estimate as the imaging experiments were performed in the absence of magnesium. It would of interest to know if presynaptic NMDARs have unique magnesium sensitivity or if voltage-dependent block can be overcome during brief train stimulation.

Our experiments were designed to demonstrate CaTs mediated by preNMDARs. In response to the reviewer’s comment regarding Mg^2+^ concentration and voltagedependent block, we performed new experiments under more physiological Mg^2+^ concentration. We found that NMDAR antagonism with D-APV also reduced presynaptic CaTs (Figure 5—figure supplement 1).